# Reliable identification of protein-protein interactions by crosslinking mass spectrometry

Swantje Lenz [1,3], Ludwig R. Sinn [1,3], Francis J. O'Reilly [1,3], Lutz Fischer[1], Fritz Wegner[1] & Juri Rappsilber [1,2 ✉]

Protein-protein interactions govern most cellular pathways and processes, and multiple technologies have emerged to systematically map them. Assessing the error of interaction networks has been a challenge. Crosslinking mass spectrometry is currently widening its scope from structural analyses of purified multi-protein complexes towards systems-wide analyses of protein-protein interactions (PPIs). Using a carefully controlled large-scale analysis of *Escherichia coli* cell lysate, we demonstrate that false-discovery rates (FDR) for PPIs identified by crosslinking mass spectrometry can be reliably estimated. We present an interaction network comprising 590 PPIs at 1% decoy-based PPI-FDR. The structural information included in this network localises the binding site of the hitherto uncharacterised protein YacL to near the DNA exit tunnel on the RNA polymerase.

[1] Bioanalytics, Institute of Biotechnology, Technische Universität Berlin, Berlin, Germany. [2] Wellcome Centre for Cell Biology, University of Edinburgh, Edinburgh, UK. [3]These authors contributed equally: Swantje Lenz, Ludwig R. Sinn, Francis J. O'Reilly. ✉email: Juri.Rappsilber@tu-berlin.de

Crosslinking mass spectrometry (Crosslinking MS) has become a key technology for understanding the architecture of multi-protein complexes by providing distance restraints between protein residues[1]. These studies are typically performed on purified complexes, but in recent years pioneering studies have used Crosslinking MS to study the topology of PPIs in more complex systems, such as cell lysates, organelles or whole cells[2–13]. Crosslinking MS is therefore emerging as a technique for mapping PPIs alongside existing tools, such as two-hybrid screens, affinity purification, proximity labelling techniques and co-fractionation studies. Importantly, Crosslinking MS studies do not require tagging of proteins and can fixate interactions inside cells prior to cell disruption. Crosslinking MS can therefore detect otherwise difficult to observe PPIs, including weak or transient interactions and interactions involving proteins that are not easily solubilised. Unlike other large-scale PPI mapping technologies, the interactions are detected between individual residues and therefore also provide information on protein complex topology.

As for any technology for mapping PPIs, the reported interactions must be reliable to be useful. Large numbers of spurious PPIs are avoided by correctly estimating FDRs and then trimming the list of reported PPIs to the desired error rate. The standard method for error estimation in classical LC-MS-based proteomics is the target-decoy approach, where a decoy database of spurious peptide sequences is included to model random identifications. This approach assumes that the rate of matches to the decoy database is an estimator of false positives (type I error rate). This target-decoy approach has been adapted for Crosslinking MS[14–18]. Recently, however, concerns have emerged regarding current FDR methods[12,19,20] and the need for improvements is recognised widely across the Crosslinking MS field[21].

Matches in Crosslinking MS are different from those in classical proteomics because two peptides are combined to make one match. This leaves two potential opportunities for a false match, which requires additional considerations when applying the target-decoy approach, such as a crosslink-specific equation for calculating FDR[15,16]. Two additional considerations have been suggested for correctly estimating errors in crosslinking-based PPI screens. The first, whether to consider crosslinks between peptides within one protein sequence (self-links, including homomeric crosslinks) separately from crosslinks between distinct protein sequences (heteromeric crosslink)[4,15]. The second, how to handle propagation of error between the different levels of information, i.e. from crosslinked spectrum matches (CSMs), to peptide pairs, to residue pairs and finally to PPIs[16]. However, both considerations have not been systematically tested and therefore they have remained controversial with no consensus emerging for if and how they should be implemented (Supplementary Table 1).

In this work, we tested different approaches for FDR estimation and demonstrated how incorrect handling of the error estimation can have huge effects on the reliability of the reported PPIs. For this, we designed a carefully controlled large-scale crosslinking study of the model organism E. coli by fractionating lysate via size exclusion chromatography (SEC), crosslinking within the individual fractions, and then pooling all fractions. Proteins that did not share the same SEC fraction could not be crosslinked and therefore reveal false PPIs, without needing to rely on decoys. We used this sample to demonstrate that self-links and heteromeric crosslinks must always be separated for FDR and that data must be merged into PPIs before correct estimation of error in crosslinking-based PPI investigations.

## Results

### Theoretical considerations on FDR estimation in crosslinking MS

Naively, FDR is estimated based on a score distribution of CSMs to the target and decoy databases, using the decoy matches as a model of random and hence false target matches. However, the size of the search space, and therefore the chance of random matching, is inherently different for heteromeric crosslinks and self-links (Fig. 1a). In our database of 4350 proteins, the chance of matching a decoy crosslink (random) within the heteromeric crosslinks is 10.6 times higher than within the self-links (Fig. 1b). Controlling FDR in the total set of CSMs, and then selecting only heteromeric matches thus enriches for false positives. This leads to a large underestimation of the error within heteromeric CSMs, which describe PPIs (Supplementary Fig. 1). Consequently, heteromeric crosslinks must be considered separately from self-links during FDR estimation.

A second consideration is that a naïve FDR for CSMs may not reflect the error among reported PPIs. When merging data from CSMs to PPIs false and true matches may behave differently and thus the relative error will change. CSMs merge into peptide pairs, peptide pairs into residue pairs and residue pairs into PPIs (see Methods). False PPIs are the result of random CSMs and thus less likely than true PPIs to be supported by multiple CSMs. Multiple true CSMs are therefore much more likely to merge into a single PPI. This leads to a change in ratio between true and false matches as one merges CSMs into PPIs. Consequently, CSMs must be merged into PPIs before FDR estimation of PPIs (Fig. 1c).

These considerations apply universally, as they are independent of crosslinker chemistry and data analysis workflow.

### Construction of a test system to investigate methods of FDR estimation in crosslinking MS

To test different approaches for FDR estimation we produced a sample for which we experimentally know a large number of the potential false PPIs. We prepared simplified cellular fractions enriched in protein complexes by separating E. coli lysate by size exclusion chromatography (Fig. 1d). The resulting 44 fractions span the molecular weight range from ~3 MDa to 150 kDa. A portion of each fraction was analysed by label-free quantitative proteomics to generate elution profiles of each protein across all 44 fractions. We identified 1926 E. coli proteins in these fractions combined. Consequently, the complexity of our sample approximates that of whole E. coli cells[22]. The abundance of the detected proteins spans six orders of magnitude, producing a challenging sample for detecting crosslinks.

The remainder of each fraction was split equally and crosslinked with BS3 or DSSO, respectively. For each crosslinker, the crosslinked fractions were then pooled and digested. The crosslinked peptides were first enriched by strong cation exchange chromatography (SCX) to enrich crosslinked peptides in nine high-salt fractions. Each high-salt SCX fraction was subsequently fractionated in a second chromatographic dimension by hydrophilic strong anion exchange chromatography (hSAX) into ten fractions. Following this extensive fractionation, the crosslinked peptides were acquired by LC-MS ($2 \times 90$ fractions, 32.5 days of mass-spectrometric acquisition per crosslinker) to generate a substantial dataset for testing FDR methods.

Proteins eluting in the same size exclusion fraction may be crosslinked in this analysis. In contrast, proteins that were not in the same fraction cannot be crosslinked together, i.e. are 'non-crosslinkable' pairs (either because they were not identified at all or below an abundance threshold (Supplementary Fig. 1). If such a non-crosslinkable protein pair is identified during data analysis, it is a false match. This experimental assessment of PPI error is independent of the target-decoy approaches and therefore offers an opportunity to benchmark target-decoy-based PPI-FDR methods.

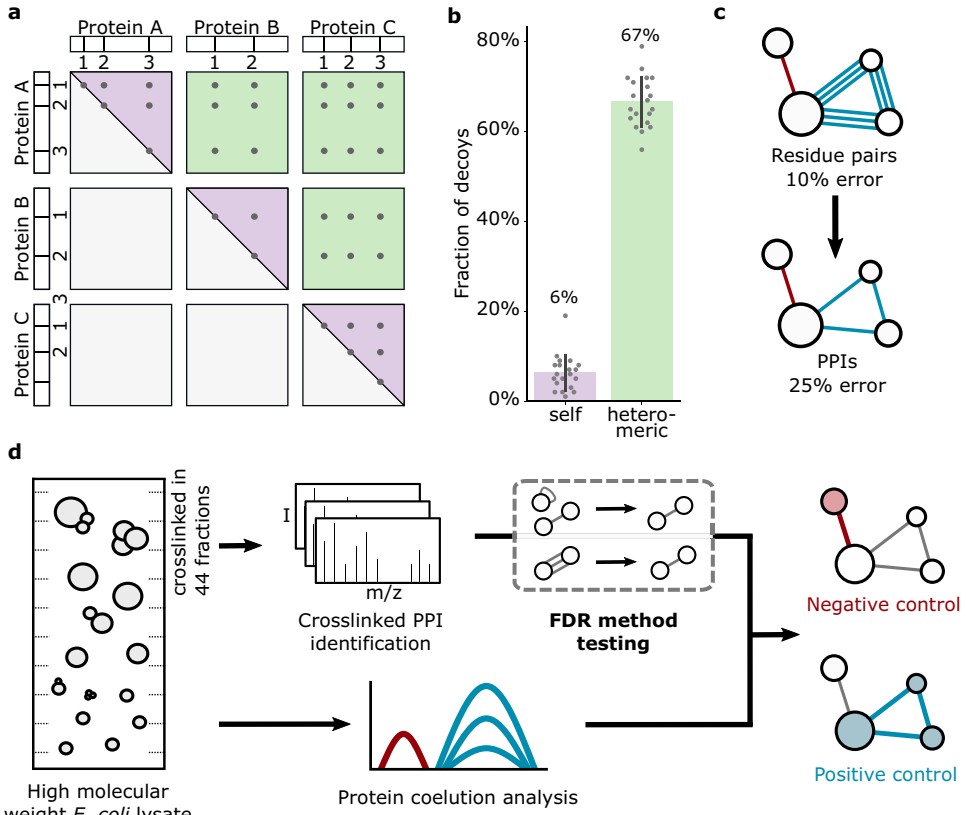

**Fig. 1 Considerations for crosslinked PPI-FDR and experimental workflow. a** For matches within the same protein sequence (with a non-directional crosslinker[17]), a crosslink from A1 to A2 is indistinguishable from A2 to A1 (theoretically possible search space shown as purple triangles). In contrast, heteromeric matches are not symmetrical, and therefore occupy a larger random space (green squares). **b** Fraction of decoys in random picks of 100 self and 100 heteromeric CSMs from the search output before any FDR filtering (random picks, n = 20, i.e. ten per crosslinker dataset). Error bars show standard deviation from the mean. Source data are provided as a Source Data file. **c** Schematic showing error increase when merging crosslinked residue pairs to PPIs. Proteins are indicated as circles; blue and red lines represent true and false linkages, respectively. **d** Experimental workflow. *E. coli* lysate was separated and crosslinked in individual high molecular weight fractions, pooled again to simulate a complex mixture and analyzed by mass spectrometry. Quantitative proteomics of uncrosslinked fractions provided protein coelution data.

**Impact of CSM-FDR estimation on the reliability of identified PPIs.** We first searched against a database comprising all (4350) *E. coli* proteins, including those not detected in our sample. Crosslinks of protein pairs defined as non-crosslinkable above were defined as false. At a naïve 5% decoy-based CSM-level FDR (not distinguishing self and heteromeric crosslinks), we identified 20,833 (5655 heteromeric) unique CSMs for BS3 and 22,296 (6923 heteromeric) unique CSMs for DSSO. We chose 5% FDR to have sufficient false identifications for precise FDR estimation at all information levels. In close agreement, our experimental control revealed that 4% of these CSMs are false (Fig. 2a, b). Note that CSMs in this manuscript refer to unique CSMs; using redundant CSMs will produce spurious FDR estimations (Supplementary Fig. 2).

However, naïve CSM-level FDR leads to many false PPIs, as our experimental control reveals. For this, the heteromeric CSMs of naïve 5% decoy-based CSM-FDR were merged into PPIs. Counting our non-crosslinkable PPIs then revealed that 36% of the reported PPIs were false in this DSSO dataset (Fig. 2a). In the BS3 dataset the results were very similar with naïve 5% decoy-based CSM-FDR leading to 35% false PPIs (Fig. 2b).

Given this large deviation between naïve decoy-based CSM-FDR and experimentally determined PPI error we sought further controls at the level of data analysis. As additional independent controls of decoy-based FDR we therefore used three entrapment database searches. First, we searched our spectra against *E. coli* sequences supplemented with the same number of human protein sequences. Second, we added the full *S. cerevisiae* proteome to the *E. coli* sequences and, finally, both databases were combined into an even larger entrapment database. Here we know any identified PPI that includes a human or yeast protein is false. According to these entrapment controls, at a naïve 5% decoy-based CSM-FDR, the PPI error reached an average of 45% (Supplementary Fig. 2). Although this corroborated the notion that naïve decoy-based CSM-FDR leads to a gross underestimation of the PPI error we devised two additional controls of decoy-based FDRs. For one, we performed searches using a fictional (wrong mass) crosslinker in addition to BS3 or DSSO, respectively. Any CSM involving this fictional crosslinker is a known false positive. While one of the two matched peptides in such a CSM might be correct, the other must be false to compensate for the false crosslinker mass when making up to the precursor mass. As a last control we searched previously high-confidence matched scans with shifted precursor masses to generate a set of false crosslinked peptides. One of the peptides constituting the original precursor could still be matched correctly. However, as the precursor mass was shifted, again, the second peptide cannot be matched correctly. So, any peptide pair match to these spectra constitutes a false positive. These controls reported a PPI error of 50 and 60%, respectively, at naïve 5% decoy-based CSM-FDR (Supplementary Fig. 2). In summary, not only the experimental but also the three entrapment and the two wrong mass controls revealed naïve decoy-based CSM-FDR to be inadequate for estimating PPI error.

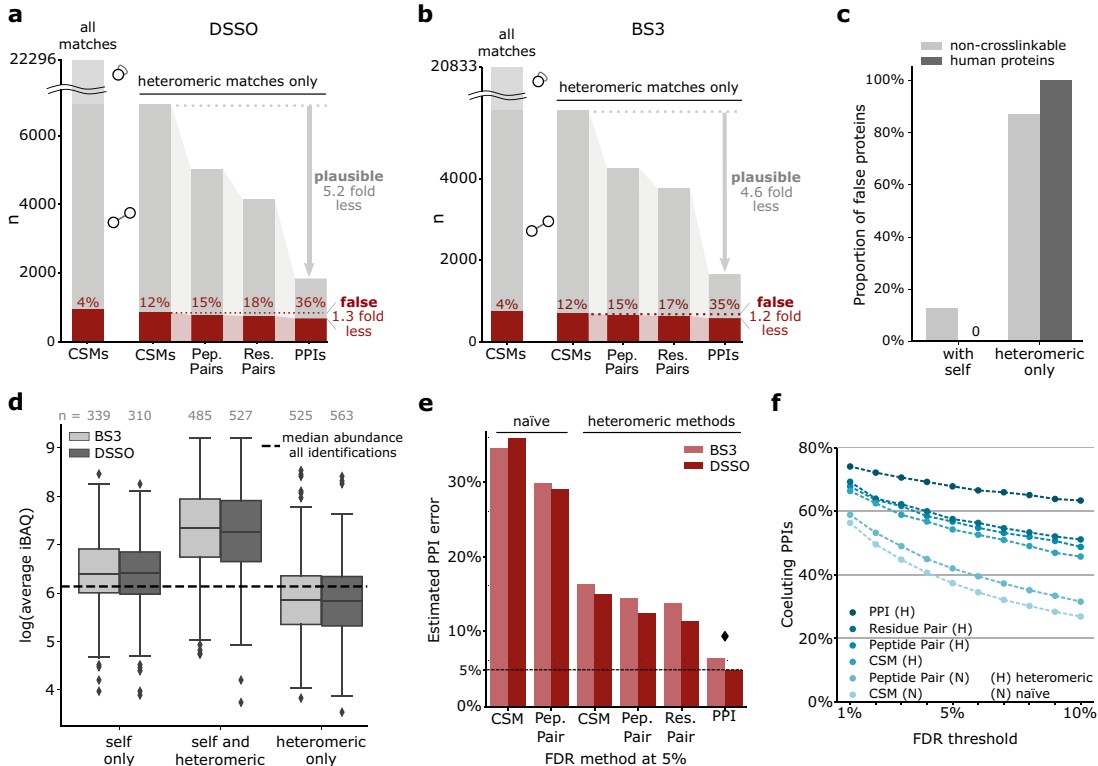

**Fig. 2 Comparative analysis of different methods of FDR estimation in crosslinking MS. a**, **b** False identifications as a function of merging heteromeric CSMs passing a naïve decoy-based CSM-FDR of 5% for **a** DSSO and **b** BS3, respectively. When merging crosslink data from CSMs to PPIs, the number of identifications decreases and the fraction of false identifications increases. CSMs rarely corroborated each other in false PPIs while plausible PPIs were supported by multiple CSMs. Heteromeric CSMs are indicted by circles connected with a straight line; self-CSMs by a circle with curved line. **c** Proportion of proteins involved in false PPIs with self-links or with only heteromeric crosslinks, of non-crosslinkable *E. coli* proteins and the entrapment database. **d** Proteins found exclusively in heteromeric PPIs had a lower abundance than all identified proteins and thus a low chance to be detected. Boxplots depict the median (middle line), upper and lower quartiles (boxes), 1.5 times of the interquartile range (whiskers) as well as outliers (single points). **e** PPI error resulting from a 5% FDR threshold of FDR approaches performed in other studies (Supplementary Table 1). Each bar is from a separate FDR calculation. Diamond denotes the method leading to the PPI error closest to 5%. **f** Fraction of protein pairs with similar elution profiles (correlation coefficient > 0.5) among the PPIs passing a given FDR threshold, applying different published FDR methods (Supplementary Table 1). Averages of BS3 and DSSO data are shown (Also presented seepearately in Supplementary Fig. 4). Source data for panels **a**, **b** and **d** are provided as a Source Data file.

Of note, in our experimental control 87% of false PPIs involved proteins that were seen only with heteromeric crosslinks, i.e. that lacked self-links (Fig. 2c). In the entrapment control this number increased to 100% (Fig. 2c). If observed at all, heteromeric-only proteins had a lower median abundance than all proteins in the sample suggesting that they are enriched in random matches (Fig. 2d). In contrast, the median abundance of proteins detected with both, self and heteromeric crosslinks, was 14.8-fold higher than the median of all identified proteins (significantly higher abundance than all identified proteins, $p < 0.0001$ using a one-sided Kolmogorov–Smirnov test) (Fig. 2d). The proportion of PPIs involving heteromeric-only proteins may thus be an indicator of reliability when evaluating published Crosslinking MS data.

**Comparative analysis of different methods of FDR estimation in crosslinking MS.** To address this inflated error of the naïve decoy-based CSM-FDR we returned to our initial theoretical considerations. Indeed, assessing heteromeric matches separately from self-matches decreased false PPIs substantially (35 to 16% and 36 to 15%, for BS3 and DSSO, respectively) (Fig. 2e). However, the error remained three times higher than the targeted 5%. We therefore also considered error propagation between information levels. As predicted, CSMs rarely corroborated each other in false PPIs while plausible PPIs were supported by multiple

CSMs (1.2 versus 4.6 for BS3, 1.3 versus 5.2 for DSSO), irrespective of the crosslinker (Fig. 2a, b). This effect was most pronounced when merging unique residue pairs into PPIs. Error control at lower information levels therefore leads to large proportions of reported PPIs being false (Fig. 2e). This also holds true for all other reporting levels (i.e. CSMs, peptide pairs and residue pairs) (Supplementary Fig. 3).

In contrast, first merging CSMs for each PPI and then assessing the FDR gave more reliable results: 6.6% and 4.9% false PPIs when applying 5% decoy-based PPI-FDR (Fig. 2e) for BS3 and DSSO, respectively. This is also supported by the other controls, which indicated an actual error close to 5% (4.8% for BS3 and 4.9% for DSSO) when applying 5% decoy-based PPI-FDR (Supplementary Fig. 2).

As a positive control, we evaluated the proportion of PPIs that were supported by correlation of protein coelution profiles (Fig. 2f, Supplementary Fig. 4). The fraction of supported PPIs was highest when using heteromeric PPI-FDR and the proportion decreased when raising the FDR threshold, as expected. The same trends are true for the alternative positive control of using interaction evidence from the STRING database (Supplementary Fig. 4).

**High-quality PPIs in *E. coli* lysate.** An FDR threshold should be chosen to meet the stringency required by the study (Supplementary Fig. 5). At a heteromeric decoy-based PPI-FDR of 5%

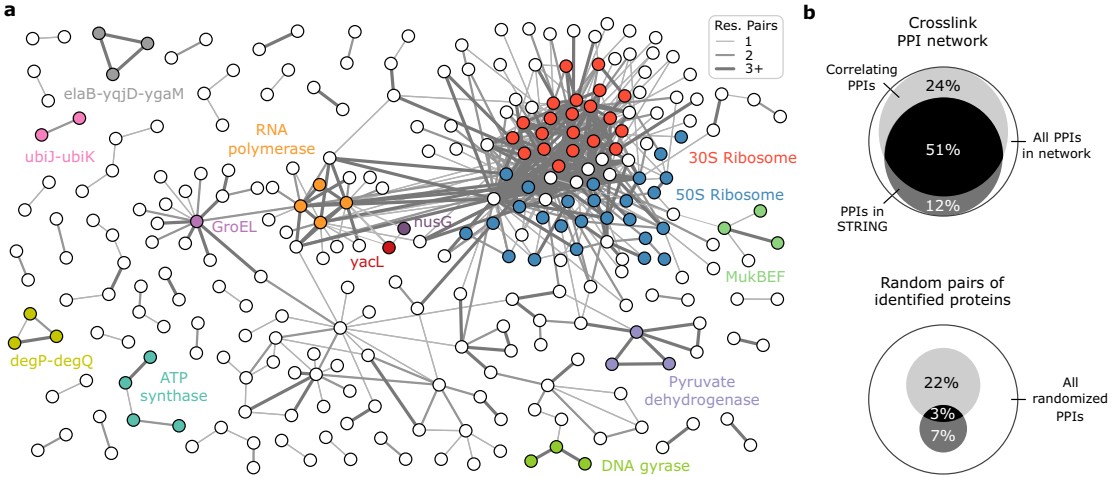

**Fig. 3 Heteromeric PPI-FDR leads to high fidelity PPI network in *E. coli* lysate. a** Crosslinking MS-derived PPI network of soluble high molecular weight *E. coli* proteome. Selected proteins (circles) and protein complexes are highlighted. The proteins AceA and TnaA were removed for clarity. **b** Characterization of the obtained PPI network in comparison to random PPIs from proteins identified in coelution data. Shown are the overlaps with STRING database and coelution data (correlation coefficient > 0.5).

applied on our data, 756 PPIs are reported, with 38 expected to be false. To focus on a high-quality subset of PPIs in the *E. coli* lysate, we applied a 1% heteromeric PPI-FDR cut-off, yielding 590 PPIs involving 308 proteins (Fig. 3a, Supplementary Data 1), connected with a total of 2539 residues pairs.

Three hundred sixty-six (62%) of these PPIs are connected by more than one residue pair (Supplementary Fig. 5). Eleven percent of the proteins found in PPIs had no self-links, but most had abundances higher than the sample median. These proteins tend to be small and thus produce few peptides, so can be difficult to observe by mass spectrometry (Supplementary Fig. 5). We found 63% (370) of the detected PPIs in the STRING database (Fig. 3b). Ninety eight percent (576) were found to be eluting in a fraction together and 68% had similar elution profiles (correlation coefficient > 0.5), suggesting that they form stable complexes (Supplementary Data 2). Ribosomal proteins had a complex elution pattern, presumably due to the presence of assembly intermediates, although many of the proteins that were found crosslinked to the ribosome are known interactors (26 of 53).

The crosslink-based PPI network included 289 protein pairs with highly similar coelution (correlation coefficient > 0.8). The majority of these were known interactions including complexes like ATP synthase, pyruvate dehydrogenase, MukBEF or DNA gyrase. The data confirmed binding of acyl carrier protein to MukBEF, and of YacG to DNA gyrase (Supplementary Fig. 6 and 7). In addition, 130 PPIs with highly similar coelution were not yet experimentally confirmed for *E. coli* K12, though 55 of these had a STRING entry based on other evidence. Novel interactions included those between the small ribosomal regulators ElaB, YgaM and YqjD, the periplasmic endoproteases DegP and DegQ, the ubiquinone biosynthesis accessory factors UbiK and UbiJ, as well as GroEL and potential substrates (Supplementary Fig. 8).

RNA polymerase (RNAP) crosslinked to 23 proteins (Fig. 4a). Previous interaction evidence was available for 20 of these, including the transcription factors RpoD and GreB, and the transcriptional regulators NusG, NusA and RapA; all crosslinks are in agreement with previously suggested binding sites (Supplementary Fig. 9). YacL, a protein of unknown function that was found to be associated with RNAP in pull-down experiments[23], crosslinked to the beta and beta' subunit of RNAP (four residue pairs), as well as to NusG (two residue pairs). It also coeluted with RNAP (correlation of 0.988) with an abundance comparable to NusG (Fig. 4b).

To confirm the interaction, we performed pulldowns using K12 strains with endogenously tagged ORFs of YacL, NusG and RpoB to carry an affinity-tag. YacL affinity-enriched RNAP and NusG (Fig. 4c) and, conversely, NusG and RpoB enriched YacL, thus confirming the association of YacL with NusG and RNAP (Supplementary Fig. 10). To further constrain the binding site of YacL on the RNAP and confirm the interaction site by use of a different crosslinker we crosslinked these affinity-enriched complexes using the photoactivatable crosslinker sulfo-SDA, which can provide a higher density of crosslinks than DSSO or BS3[24]. Sulfo-SDA crosslinking either of the three affinity-enriched proteins confirmed the direct binding of YacL to the RNAP and NusG with a combined 14 unique residue pairs (Supplementary Fig. 10 and Supplementary Data 3-5). The total of 20 residue pairs from DSSO, BS3 and sulfo-SDA thus constrain the binding site of YacL to RNAP and NusG. Sixteen of these residue pairs were between our I-TASSER model of YacL and regions of RNAP-NusG included in the solved structure (PDB 6C6U [https://doi.org/10.2210/pdb6c6u/pdb]). These were used in DisVis to calculate the accessible interaction space and localize YacL on RNAP next to NusG at the DNA exit site (Fig. 4d).

## Discussion

There have been several recent advances in enrichment and detection of crosslinked peptides by Crosslinking MS that suggest it will soon be able to map large portions of the cellular inter-actome in a single experiment[3,13,25]. This will open the door to detecting changes in these interactomes in different cellular states by quantification of the abundances of the detected crosslinks[2,26]. All of these advances require correctly controlled FDR to produce results that can be relied upon.

In previous studies, the quality of identified crosslinks was assessed by measuring inter-residue distances in known protein structures. However, for proteome-wide crosslinking studies, this approach is inherently biased towards true interactions as they are likely to be enriched in known complexes[27]. The majority of random PPIs are neglected by this FDR evaluation method, making this approach completely inadequate for reliable PPI error estimation.

In this work, we experimentally demonstrated that Crosslinking MS can reliably identify PPIs using the target-decoy approach as a quantitative error metric. Decoys are only a model of false positives with a number of underlying

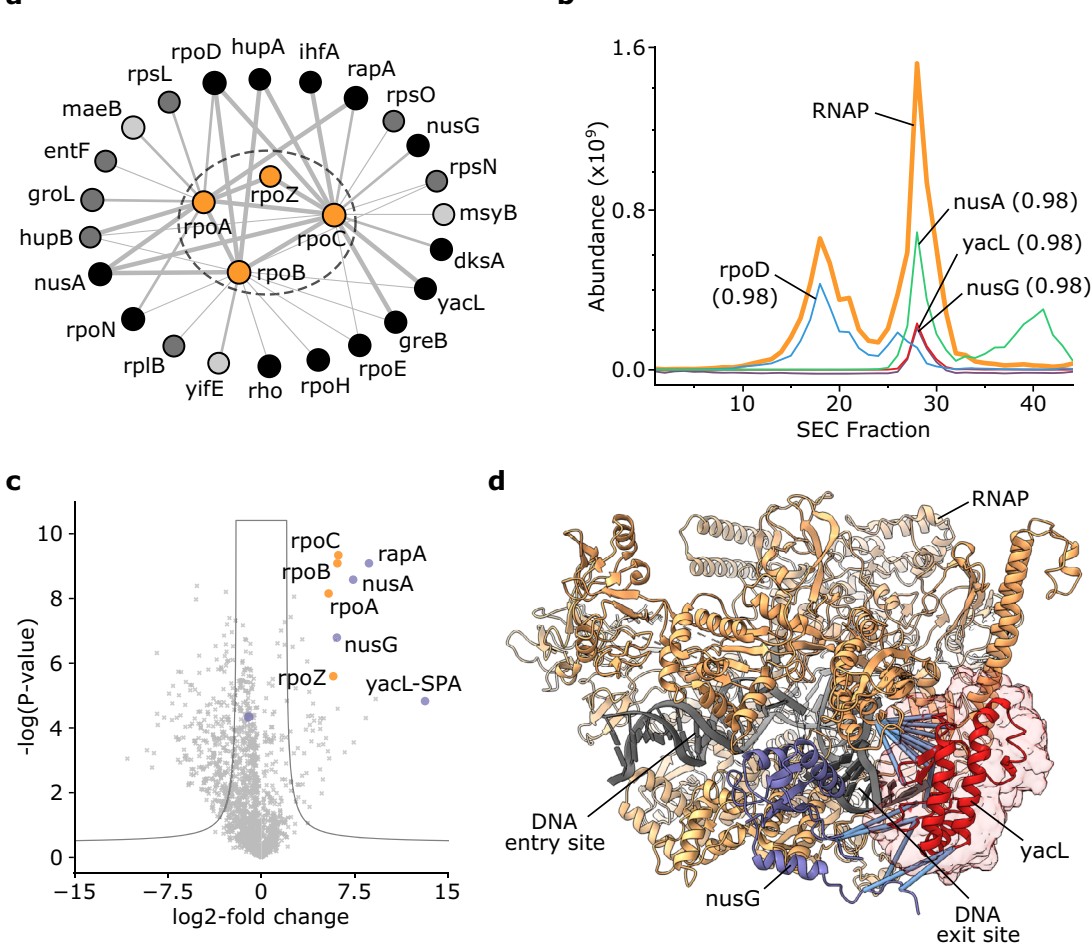

**Fig. 4 The uncharacterised protein YacL binds to RNA polymerase. a** PPI subnetwork of RNAP. Line thickness between proteins (circles) increases with frequency of observed crosslinks (i.e. one, two, three and more crosslinked residue pairs). Colour scheme for RNAP binders: light grey = coelution correlation > 0.5, dark grey = STRING database combined score ≥ 150, black = both of the previous categories. **b** Elution traces of RNAP (average abundance of its subunits) and selected RNAP binders with their minimal elution correlation coefficient to any RNAP constituent. **c** Volcano plot showing the affinity enrichment of SPA-tagged YacL, which co-enriches RNAP (orange) and a number of proteins found crosslinked to the RNAP in the YacL affinity-enrichment experiment (violet). **d** RNAP with bound NusG (PDB 6C6U [10.2210/pdb6c6u/pdb]) and the region of Crosslinking MS-defined accessible interaction space with 14 satisfied restraints for YacL (I-TASSER model, placed for visualisation purposes) highlighted. Crosslinks between YacL and NusG or RNAP are highlighted in blue.

assumptions[16] and they cannot model false positives that do not arise from spectral matching, such as peptides non-covalently associating during LC-MS[28]. Considering these caveats, it is reassuring that our four different controls closely agree with the outcome of the target-decoy approach. This negates the need for any additional heuristics suggested by others[27]. We showed that the target-decoy approach requires separating self and heteromeric crosslinks and that error should be estimated for the information level that is being reported. For example, when reporting residue pairs for structural analyses of individual protein complexes, residue pair-level FDR should be applied. However, when reporting PPIs, CSMs need to be merged to PPI level prior to FDR estimation. Other ways of merging CSM scores into PPI scores from the one we use here are possible. However, for accurate PPI error estimation, these methods would need to adhere to the two fundamental considerations. These concepts were implemented in our open-source FDR estimation software tool, xiFDR v2.0, which is crosslink search software independent. The large dataset presented here, with its internal controls, will allow testing of other aspects of the Crosslinking MS workflow in the future.

Correctly controlled error is an important element of any discovery-based technology. This remains a challenge even in well-established PPI mapping technologies including two-hybrid and affinity purification studies. Crosslinking MS for mapping PPIs now has a reliable FDR estimation procedure. This is an essential prerequisite for this technology to bridge the gap between structural studies and systems biology by reliably revealing topologies of PPIs in their native environments.

## Methods

**Materials**. Unless otherwise stated, reagents were purchased in the highest quality available from Sigma (now Merck, Darmstadt, Germany). Empore 3M C18-Material for LC-MS sample cleanup was from Sigma (St. Louis, MO, USA), glycerol from Carl Roth (Karlsruhe, Germany). The BS3 (bis (sulfosuccinimidyl) suberate) and sulfo-SDA (sulfosuccinimidyl 4,4'-azipentanoate) crosslinkers were supplied by Pierce Biotechnology (Thermo Fisher Scientific, Waltham, MA, USA) and the DSSO (disuccinimidyl sulfoxide) crosslinker from Cayman Chemical (Ann Arbor, MI, USA).

**Biomass production**. A single clone of *Escherichia coli* K12 strain (BW25113 purchased from DSMZ, Germany; https://www.dsmz.de/) grown on Agar plates was selected for inoculation of lysogeny broth (LB)-media. A preculture aliquot was used to start fermentation in a Biostat A plus bioreactor (Sartorius, Göttingen,

Germany) in LB medium with 0.5% (w/v) glucose and at 37 °C. The pH and dissolved oxygen were monitored and adjusted by the addition of sodium hydroxide/phosphoric acid or stir speed control, respectively. Overall growth was monitored by optical density measurements at 600 nm. When the culture reached an optical density of 10, the fermentation was stopped and the culture rapidly cooled in stirred ice water followed by harvesting the biomass by centrifugation at $5000 \times g$, 4 °C for 15 min. Cell pellets were stored at −80 °C after washing with PBS and snap-freezing in liquid nitrogen.

For pull-down experiments, *Escherichia coli* K12 strains with endogenously C-terminal SPA-tagged rpoB, nusG and yacL (purchased from Horizon, Cambridge, United Kingdom, https://horizondiscovery.com/) were plated according to distributor's instructions. A single clone of each strain was selected for genetic validation and subsequent starter cultures. Gene sequences were validated by PCR using primers hybridizing upstream of each open reading frame of interest and within the SPA-tag sequence (Supplementary Data 6). With the exception of the yacL-ORF which had a non-silent point mutation (Q118L), all protein- and tag-coding sequences were correct. Production cultures were inoculated into terrific broth medium and cultivated at 32 °C in baffled flasks until late log-phase. Biomass was harvested and stored after snap-freezing as described above.

**Cell lysis and high-molecular-weight proteome fractionation by size exclusion chromatography (SEC)**. Cell pellets were suspended at 0.2 g wet-mass per ml in ice-cold lysis buffer (50 mM Hepes pH 7.2 at RT, 50 mM KCl, 10 mM NaCl, 1.5 mM MgCl₂, 5% (v/v) glycerol, 1 mM dithiothreitol (DTT), spatula tip of chicken egg white lysozyme (Sigma, St. Louis, MO, USA)). Cells were lysed by sonication on ice. Prior to sonication cOmplete EDTA-free protease-inhibitor cocktail (Roche, Basel, Switzerland) was added according to the manufacturer's instructions. After sonication, 125 units of Benzonase (Merck, Darmstadt, Germany) were added. Subsequently, the lysate was cleared of cellular debris by centrifugation for 15 min at 4 °C and $15,000 \times g$. DTT was added again to 2 mM. This cleared lysate was subjected to ultracentrifugation using a 70 Ti fixed-angle rotor for 1 h at $106,000 \times g$ and 4 °C. Then, the supernatant was concentrated using ultrafiltration with Amicon spin filters (15 kDa molecular weight cut-off) to reach a total protein concentration of 10 mg/ml, as judged by microBCA assay (Thermo Fisher Scientific, Waltham, MA, USA). Aggregates were removed by centrifugation for 5 min at $16,900 \times g$ and 4 °C. Two milligrams of soluble high molecular weight proteome was loaded onto a BioSep SEC-S4000 column (600 × 7.8 mm, pore size 500 Å, particle size 5 μm, Phenomenex, CA, USA) and fractionated at 200 μl/min flow rate and 4 °C while collecting fractions of 200 μl over the separation range from ~3 MDa to 150 kDa (as judged by Gel filtration calibration kit (HMW), GE Healthcare).

**Affinity-pulldowns of RNA polymerase constituents and binders**. Cells were lysed by sonication identically to the protocol described above. The supernatants from centrifugation for 1 h at 4 °C and $20,000 \times g$ were incubated with washed Anti-FLAG M2 agarose beads (Sigma, St. Louis, MO, USA) on a vertical rotator for 2 h at 4 °C, according to the specifications of the manufacturer. Supernatants after incubation were discarded and beads washed twice with wash buffer (10 mM Tris*HCl pH 7.4 at RT, 100 mM NaCl, 10% (v/v) glycerol) and once with modified lysis buffer (50 mM Hepes pH 7.2 at RT, 50 mM KCl, 10 mM NaCl, 1.5 mM MgCl₂, 5% (v/v) glycerol). M2 beads from replica pulldowns were pooled in a single tube and again resuspended in modified lysis buffer. TEV protease (Sigma, St. Louis, MO, USA) was added >0.5 U/μl M2 beads and the protein complexes of interest eluted over 1 h at 16 °C with gentle agitation. Aliquots of cleared supernatants and eluates from TEV cleavage were collected and processed as described below.

**Sample preparation for LC-MS protein identification with non-crosslinked samples**. For protein identification from SEC fractionation, aliquots (40 μl) of each fraction were precipitated by adding four volumes of cold acetone followed by an incubation at −20 °C overnight. Pellets were collected by centrifugation and supernatants discarded. Protein pellets were air-dried and subsequently solubilized using 6 M urea, 2 M thiourea, 100 mM ABC (ammonium bicarbonate). Derivatization was accomplished by incubating for 30 min at RT with 10 mM DTT followed by 20 mM IAA (iodoacetamide) for 30 min in the dark at RT, respectively. Proteases were added to the samples: LysC (1:100 (m/m)) at 37 °C for 4.5 h at 37 °C, followed by diluting 1:5 with 100 mM ABC and continued with trypsin (1:25 (m/m)) at 37 °C for 16 h. The reactions were stopped by adding TFA (trifluoroacetic acid) to a pH of 2–3. Subsequently, sample cleanup following the Stage-tip protocol was performed and samples were stored at −20 °C until LC-MS acquisition. Samples from pulldowns were processed similarly with the following changes: use of 8 M urea, 100 mM ammonium bicarbonate for solubilization; blocking of reduced cysteines with 30 mM IAA; digestion with trypsin (ca. 1:50 (m/m)).

**LC-MS protein identification with non-crosslinked samples**. Protein identifications in SEC fractions and from pull-down experiments via LC-MS were conducted using a Q Exactive HF mass spectrometer (Thermo Fisher Scientific, Bremen, Germany) coupled to an Ultimate 3000 RSLC nano system (Dionex, Thermo Fisher Scientific, Sunnyvale, USA), operated under Tune 2.9, SII for Xcalibur 1.4 and Xcalibur 4.1. 0.1% (v/v) formic acid and 80% (v/v) acetonitrile,

0.1% (v/v) formic acid served as mobile phases A and B, respectively. Samples were loaded in 1.6% acetonitrile, 0.1% formic acid on an Easy-Spray column (C18, 50 cm, 75 μm ID, 2 μm particle size, 100 Å pore size) operated at 45 °C and running with 300 nl/min flow. Peptides were eluted with the following gradient: 2 to 6% buffer B in 1 min, 6 to 10% B in 2 min, 10 to 30%B in 37 min, 30 to 35% in 5 min followed by 35 to 45%B in 2 min. Then, the column was set to washing conditions within 1.5 min to 90% buffer B and flushed for another 5 min. For the mass spectrometer the following settings were used: MS1 scans resolution 120,000, AGC (automatic gain control) target $3 \times 10^6$, maximum injection time 50 ms, scan range from 350 to 1600 $m/z$. The ten most abundant precursor ions with z = 2–6, passing the peptide match filter ("preferred") were selected for HCD (higher-energy collisional dissociation) fragmentation employing stepped normalized collision energies (29 ± 2). The quadrupole isolation window was set to 1.6 $m/z$. Minimum AGC target was $2.5 \times 10^4$, maximum injection time was 80 ms. Fragment ion scans were recorded with a resolution of 15,000, AGC target set to $1 \times 10^5$, scanning with a fixed first mass of 100 $m/z$. Dynamic exclusion was enabled for 30 s after a single count and included isotopes. Each LC-MS acquisition took 75 min.

**Quantitative proteomics database search**. Raw data from bottoms-up proteomics experiments were processed using MaxQuant[29] version 1.6.0.16 operated under default settings (fully tryptic digestion with two missed cleavages maximum; up to five variable modifications per peptide (oxidised methionine and acetylated protein N-termini), MS1 match tolerance 20 ppm (first search)/4.5 ppm (main search), MS/MS match tolerance 20 ppm); carbamidomethylation of cysteine set as fixed modification; 1% PSM and protein group FDR). Each SEC fraction or pull-down replica injection was treated as an individual experiment. Quantitation by iBAQ[30] requiring a minimum of two peptides (unique + razor) and matching between runs were enabled. For data from pull-down experiments, label-free quantitation was enabled with default settings (LFQ minimum ratio count of 2, Fast LFQ enabled, minimum number/average of neighbour 2/6, stabilize large LFQ ratios and requirement for MS2 for LFQ comparisons enabled). Supernatant samples from cell lysis were included to increase absolute protein identifications via the matching between runs feature. The database used was the Uniprot curated reference proteome UP000000625 with two unreviewed entries removed summing to a total 4350 proteins (retrieved on 04/08/2019).

Protein enrichment from pull-down experiments was assessed using Perseus[31] version 1.5.6.0. Proteins identified by site only, reverse hits and contaminants were filtered out. LFQ protein quantitation data was log2-transformed and filtered to contain three valid values in at least one experiment (e.g. in any TEV eluate). Missing values were imputed on the total matrix with default settings (width: 0.3, downshift 1.8). Volcano plots comparing TEV eluates of targeted affinity enrichment with K12 wildtype mock enrichment were created using a two-sided, two-sample *t*-test with 1% FDR and an artificial variance S0 of 2. For high-resolution figures, the matrix and the cut-off curve were exported to reproduce the plots in python 3.7 with pandas 0.24.2 using the seaborn 0.9.0 package.

**Protein crosslinking, digestion and sample cleanup of SEC fractions**. The remaining parts of the SEC fractions (160 μl, see above) were split into two 75 μl aliquots, for the two crosslinking reactions, and adjusted to 97.5 μl with 1× SEC-Running buffer. Crosslinker stock solutions were prepared freshly at 30 mM in water free DMF. Crosslinking of the fractions was initiated by quickly mixing each sample with 2.5 μl crosslinker stock to a final concentration of 0.75 mM crosslinker. The crosslinking reaction was incubated for 2 h on ice before quenching with ABC at 50 mM and further incubation for 30 min on ice. The crosslinked samples were acetone-precipitated at −20 °C overnight (see above). Protein was solubilized in 6 M Urea, 2 M Thiourea, 100 mM ABC. For sample reduction and alkylation, 10 mM DTT for 30 min at RT and 20 mM IAA for 30 min at RT in the dark were employed. For sample proteolysis, LysC was added at 1:100 (m/m) ratio and incubated for 4 h at 37 °C. Upon 1:5 dilution with 100 mM ABC, Trypsin was added to the sample (1:25 (m/m)) and digestion continued for 16 h at 37 °C followed by stopping via addition of TFA to a pH of 2–3. The digests were desalted using SPE cartridges following the manufacturer's instruction and eluates dried, aliquoted and stored at −20 °C until further use.

**Multidimensional offline fractionation of crosslinked peptide samples**. All crosslinked peptide pools were fractionated using an Äkta pure system (GE Healthcare, Chicago, IL, USA) employing a PolySulfoethyl A SCX column (100 × 2.1 mm, 300 Å, 3 μm) equipped with a guard column of identical stationary phase (10 × 2.0 mm) (PolyLC, Columbia, MD, USA) running at 0.2 ml/min for the first separation dimension. Here, mobile phase A consisted of 10 mM KH₂PO₄ pH 3.0, 30% ACN while mobile phase B contained 1 M KCl in addition. The system was kept at 21 °C throughout the fractionation. Dried digestion aliquots of 400 ug peptides were dissolved in mobile phase A. Upon injection, peptides were eluted isocratically for 2 min followed by an exponential gradient up to 700 mM KCl with following steps: 12 min to 12.7%, followed by 1-min steps to 14.5, 16.3, 18.8, 23.0, 30.0, 40.0, 70.0% B. Fractions of 200 μl size were collected over the elution range. The same nine high-salt fractions from five replica SCX runs were pooled for desalting using Stage-tips. Dried Stage-tip eluates of each individual SCX fraction were then subjected to the second dimension offline fractionation by hSAX

chromatography. Here, a Dionex IonPac AS-24 hSAX column (250 × 2.0 mm) with an AG-24 guard column (Thermo Fisher Scientific, Dreieich, Germany) were used on Äkta pure system (see above). Mobile phase A consisted of 20 mM Tris*HCl pH 8.0 with mobile phase B containing 1 M NaCl in addition. The system was kept at 15 °C for these experiments. Samples were eluted from Stage-tips, dried and resuspended in mobile phase A. Again, peptides were loaded under isocratic conditions for 3 min, and then eluted by an exponential gradient with the following steps 1.8, 3.5, 5.3, 7.1, 9.1, 11.2, 13.5, 16.3, 19.7, 24.1, 30.2, 38.8, 51.5, 70.6, 100% B lasting for one minute each. Fractions of 150 µl size were collected throughout the elution phase. Adjacent fractions were pooled to give ten pools in total (fractions 3–6/7–14/15–17/18–19/20–21/22–23/24–25/26–27/29–29/30–35), that were desalted using Stage-tips.

**Protein crosslinking, digestion and crosslink enrichment of pull-down eluates.** The remaining pull-down eluate fractions (minus aliquots for protein identification, see above) were split into five fractions. The heterobifunctional photo-activatable crosslinker sulfo-SDA was dissolved in modified lysis buffer (50 mM Hepes pH 7.2 at RT, 50 mM KCl, 10 mM NaCl, 1.5 mM MgCl$_2$, 5% (v/v) glycerol) and immediately added to the samples at 50, 100, 250, 500 and 1000 µM. The crosslinking reaction proceeded in the dark for 2 h on ice. UV-crosslinking was achieved by irradiation with a high-power UV-A LED laser (LuxiGenTM) at 365 nm for 15 s at one Ampere[32]. Samples were frozen and stored at −20 °C. Next, the samples were denatured by adding solid urea to give an 8 M solution, reduced using DTT at 10 mM following incubation at RT for 30 min and derivatized at 30 mM IAA over 20 min at RT and in the dark. LysC protease was added (protease:protein ratio ca. 1:100 (m/m)) and the samples digested for 4 h at 37 °C. Then, the samples were diluted 1:5 with 100 mM ABC and trypsin was added at a ratio of ~1:50 (m/m). Digestion progressed for 16 h at 37 °C until stopping with TFA. Digests were cleaned up using C18 StageTips.

Eluted peptides were fractionated using a Superdex Peptide 3.2/300 column (GE Healthcare, Chicago, IL, USA) at a flow rate of 10 µl min−1 using 30% (v/v) acetonitrile and 0.1 % (v/v) trifluoroacetic acid as mobile phase[33]. Early 50-µl fractions were collected, dried and stored at −20 °C prior to LC-MS analysis.

**LC-MS for crosslink identification.** LC-MS analysis of crosslinked peptides derived from the SEC-separated *E. coli* proteome and multidimensional fractionation was performed using a Q Exactive HF mass spectrometer (Thermo Fisher Scientific, Bremen, Germany) coupled to an Ultimate 3000 RSLC nano system (Dionex, Thermo Fisher Scientific, Sunnyvale, USA), operated under Tune 2.11, SII for Xcalibur 1.5 and Xcalibur 4.2. Mobile phases A and B consisted of 0.1% (v/v) formic acid and 80% (v/v) acetonitrile, 0.1% (v/v) formic acid, respectively. Samples were loaded in 1.6% acetonitrile, 0.1% formic acid on an Easy-Spray column (C18, 50 cm, 75 µm ID, 2 µm particle size, 100 Å pore size) running at 300 nl/min flow and kept at 45 °C. Analytes were eluted with the following gradient: 2 to 7.5% buffer B in 5 min, followed by a linear 80-min gradient of 7.5 to 42.5% and an increase to 50% B over 2.5 min. Then, the column was set to washing conditions within 2.5 min to 95% buffer B and flushed for another 5 min. The mass-spectrometric settings for MS1 scans used were: resolution set to 120,000, AGC of 3 × 10⁶, maximum injection time of 50 ms, scanning from 400–1450 m/z in profile mode. The ten most intense precursor ions that passed the peptide match filter ("preferred") and with z = 3–6 were isolated using a 1.4 m/z window and fragmented by HCD using in-house optimized stepped normalized collision energies (BS3: 30 ± 6; DSSO: 24 ± 6). Fragment ion scans were acquired at a resolution of 60,000, AGC of 5 × 10⁴, maximum injection time of 120 ms scanning from 200–2000 m/z, underfill ratio set to 1%. Dynamic exclusion was enabled for 30 s (including isotopes). In-source-CID was enabled at 15 eV to minimize gas-phase associated peptides[28]. Each LC-MS run took 120 min.

For LC-MS/MS analysis of sulfo-SDA crosslinked samples, we used an Orbitrap Fusion Lumos Tribrid mass spectrometer (Thermo Fisher Scientific, Germany) connected to an Ultimate 3000 RSLCnano system (Dionex, Thermo Fisher Scientific, Germany), which were operated under Tune 3.4, SII for Xcalibur 1.6 and Xcalibur 4.4. Fractions from SEC were resuspended in 1.6% acetonitrile 0.1% formic acid and loaded onto an EASY-Spray column of 50 cm length (Thermo Scientific) running at 300 nl/min. Gradient elution using water with 0.1% formic acid and 80% acetonitrile with 0.1% formic acid was accomplished using optimised gradients for each SEC fraction (from 2–18% mobile phase B to 37.5-46.5% over 90 min, followed by a linear increase to 45–55 and 95% over 2.5 min each). Each fraction was analysed in duplicate. The settings of the mass spectrometer were as follows: Data-dependent mode with 2.5s-Top-speed setting; MS1 scan in the Orbitrap at 120,000 resolution over 400 to 1500 m/z with 250% normalized AGC target; MS2 scan trigger only on precursors with z = 3–7+, AGC target set to "standard", maximum injection time set to "dynamic"; fragmentation by HCD employing a decision tree logic with optimised collision energies[34,35]; MS2 scan in the Orbitrap at resolution of 60,000; dynamic exclusion was enabled upon a single observation for 60 s.

**Crosslink database search for BS3 and DSSO.** Raw data from mass spectrometry were processed using msConvert (version 3.0.11729)[36] including denoising (top 20 peaks in 100 m/z bins) and conversion to mgf-file format. Precursor masses were

re-calibrated to account for mass shifts during measurement. Obtained peak files were analysed using xiSEARCH 1.6.746[5] with the following settings: MS1/MS2 error tolerances 3 and 5 ppm, allowing up to two missing isotope peaks[37], tryptic digestion specificity with up to two missed cleavages, carbamidomethylation on cysteine as fixed and oxidation on methionine as variable modification, losses: –CH$_3$SOH/–H$_2$O/–NH$_3$, crosslinker BS3 (138.06807 Da linkage mass) or DSSO (158.0037648 Da linkage mass) with variable crosslinker modifications on linear peptides ("BS3-NH2" 155.09463 Da, "BS3-OH" 156.07864 Da, "DSSO-NH2" 175.03031 Da, "DSSO-OH" 176.01433 Da). xiSEARCH algorithms are identical for both crosslinkers (BS3 and DSSO). For samples crosslinked with DSSO, additional loss masses for crosslinker-containing ions were defined accounting for its cleavability ("A" 54.01056 Da, "S" 103.99320 Da, "T" 85.98264). Matches were not filtered for having DSSO-specific signature peaks. Crosslink sites for both reagents were allowed for side chains of Lys, Tyr, Ser, Thr and the protein N-terminus. Note that we included a non-covalent crosslinker with a mass of zero to flag spectra potentially arising from gas-phase associated peptides. These spectra were removed prior to false-discovery-rate (FDR) estimation[28].

As for the non-crosslinked samples, the full *E. coli* proteome of 4350 proteins was used. For the entrapment database control the database was extended by three different entrapment databases (see below). For the final PPI network, the search database was reduced to only proteins identified in our 44 SEC fractions, to reduce noise in the database. Decoys were generated for all searches, including the entrapment database. For this, protein sequences were reversed and for each decoy protein the enzyme specific amino acids were swapped with their preceding amino acid[29].

**FDR calculation for BS3 and DSSO datasets.** Results were filtered prior to FDR to crosslinked peptide matches having a minimum of three matched fragments per peptide, a delta score of 15% of the match score and a peptide length of at least six amino acids. Additionally, identifications ambiguously matching to two proteins or more were removed. FDR was calculated based on decoy matches by xiFDR (version 2.0dev) using Eq. (1):[16]

$$FDR = \frac{TD - DD}{TT} \qquad (1)$$

Depending on the experiment, FDR was employed on different result levels (CSM, peptide pair, residue pair or protein pair) with defined thresholds. Scores of higher levels were calculated as described by Fischer and Rappsilber[16] using Eq. (2):

$$Score_{higher\ level} = \sqrt{\Sigma\ (Score_{lower\ level})^2} \qquad (2)$$

FDR was solely calculated based on that score, no further improvement by other information was done at this point. To account for the improvement of the identification by prefiltering on lower levels[16], the same threshold was employed on each of the lower levels. Self- and heteromeric crosslinks were handled together or separately by enabling/disabling the grouping option.

For the final PPI network, BS3 and DSSO PPIs were separately filtered to 1% heteromeric PPI-FDR. As the score cut-offs differed between the two datasets, the scores from each dataset were first normalized (i.e. the local FDR was used as a normalized score) to range between 0 and 1. Subsequently, the two tables were concatenated and the FDR calculated again as described above and filtered to 1% FDR.

**Non-crosslinkable control.** Due to the high sensitivity of mass spectrometry we identified a long list of proteins in each SEC fraction. Theoretically, all of the proteins in a fraction could be crosslinked. In practice, however, even if this were the case, we could not detect all these crosslinks because many would be below our detection limit. We therefore set out to heuristically determine the detection limit in our analysis. For this, the iBAQ values were determined across all SEC fractions of all proteins that were part of an identified heteromeric peptide pair, at a generous 10% heteromeric peptide-pair FDR. For each identified pair of proteins, the respective iBAQ pairs were determined across all SEC fractions (note that iBAQ could be zero if a protein was not identified in a given fraction). Looking into each fraction the lower of the two iBAQ values was kept. The maximum of this distribution over all fractions was then taken, called "best lower iBAQ" of a protein pair. We assumed this to be the appropriate abundance estimate for a protein pair and therefore the best estimate of the chance for this pair to be observed in our experiment as crosslinked. The question now is what abundance is sufficient. As a heuristic, we removed the lower 5% iBAQ values (iBAQ of 4.3E6). This removed very few (169, 7%) of our identified protein pairs (n.b. at a very loose FDR threshold), i.e. did not change much the outcome of our identification data by generating false negatives.

For any identified protein pair to be considered plausible, both proteins had to be found in at least one SEC fraction together with individual iBAQ values above our iBAQ threshold (iBAQ of 4.3E6). Otherwise, they were defined as non-crosslinkable. 544,274 (6% of all theoretically possible PPIs in the *E. coli* proteome of 4350 proteins) are defined as plausible (Supplementary Data 7), while 8,914,801 (94%) PPIs are non-crosslinkable. Note that unlike an error control using an entrapment database during search, only proteins that could make up the sample were considered.

Additionally, the difference in the sizes of the false and plausible search spaces needs to be taken into account for error estimation, i.e. the different number of possible tryptic peptides. While all matches in the false search space are false by

definition, some matches in the plausible search space will also be random. To account for these, the Lysine/Arginine content of proteins in the respective groups was used as an estimate for the number of possible peptides and the observed error is calculated with Eq. (3):

$$\text{Error}_{\text{PPI,non-crosslinkable}} = \frac{n_{\text{false}}}{n_{\text{plausible}} + n_{\text{false}}} \cdot \frac{KR_{\text{plausible}} + KR_{\text{false}}}{KR_{\text{false}}} \quad (3)$$

$$= \frac{n_{\text{false}}}{n_{\text{plausible}} + n_{\text{false}}} \cdot 1.09$$

where $n_{\text{false}}$ is the number of PPIs defined as "non-crosslinkable", $n_{\text{plausible}}$ the number of PPIs that are plausible, both after passing the respective FDR calculation. $KR_{\text{plausible}}$ and $KR_{\text{false}}$ are the sums of Lysines and Arginines in the proteins of the plausible or false interactions, respectively. Therefore, the Lysine and Arginine normalisation factor, here 1.09, is database specific.

**Entrapment database control calculation**. As a second control, the error of matched PPIs was estimated based on known wrong matches to three entrapment databases of different sizes. For one, the same number (4350) of human proteins of similar size was added by sampling a human protein similar in Lysine and Arginine content for each E. coli protein. As a second entrapment database, the full S. cerevisiae proteome was added. Finally, both databases were combined for a third entrapment database.

PPIs were defined as false if one or more proteins in the PPI was a human or yeast protein. Additionally, the difference in entrapment and possible search space has to be taken into account, similar to the approach for the non-crosslinkable control, following Eq. (4):

$$\text{Error}_{\text{PPI,entrapment}} = \frac{n_{\text{entrapment}}}{n_{\text{E.coli}} + n_{\text{entrapment}}} \cdot \frac{KR_{\text{E.coli}} + KR_{\text{entrapment}}}{KR_{\text{entrapment}}} \quad (4)$$

As expected from doubling the original database size by adding an entrapment database of equal size, the search space normalisation approximates to 2 (1.998) for the human entrapment database. For the yeast proteome and human proteins and yeast proteome databases the Arginine and Lysine normalization factors are 1.19 and 1.16, respectively.

**Wrong crosslinker control**. As a third control we performed searches using a wrong mass crosslinker in addition to BS3 or DSSO, respectively. Both crosslinker masses were reduced by 28.031 Dalton. Note that for these searches, DSSO was treated as non-cleavable. Wrong mass matches were treated separately, i.e. the same PPI matched to correct and wrong crosslinker appeared twice in the results. The error was normalized by a factor of 2 (see above).

**Wrong precursor mass control**. Spectra passing a 1% heteromeric CSM-FDR were extracted. For these spectra, the precursor mass was downshifted by 28.031 and 42.047 Dalton (corresponding to the mass of two or three methylations). Correct and shifted spectra were searched together, every match to a known wrong spectrum counted as wrong. Here, the error was not corrected as we assume the unknown wrong matches in the correct mass spectra to be only at 1% based on the FDR employed before.

**Correlation of protein elution profiles**. Proteins were quantified in each SEC fraction as described above. iBAQ values for each protein were normalized by the maximum of the respective protein over the course of fractionation, leading to normalized abundance values between 1 and 0. For each combination of proteins, elution peaks were detected via the scipy python package (1.4.1). In an elution window of 7 or more fractions, the abundances of the proteins were correlated (Pearson). PPIs with elution profiles with a correlation coefficient >0.5 were counted as having similar elution profiles. Code was written in python 3.7.

**PPI network comparison with STRING database**. For all E. coli K12 proteins identified in quantitative proteomics experiments, interaction evidence from the STRING database v10.5[38] was used (scores ranging from 0 to 1000, retrieved from https://string-db.org on 12/19/18). PPIs were accounted as known if the STRING combined score was equal or higher than 150. PPIs were defined as lacking experimental evidence when the STRING experimental score was lower than 150. Note that STRING defines 150 as the lowest cut-off in favour of an interaction.

**Plotting protein elution profiles**. For the creation of protein elution profiles, fraction-wise iBAQ intensities for each protein from the MaxQuant search were used (see above), hereinafter referred to as abundance. Individual proteins are represented by their gene names while protein complexes, when shown in the figures, are labelled with their respective complex name. Abundance values for protein complexes were averaged for all components as listed in EcoCyc[39], (retrieved from https://ecocyc.org/ on 9/25/18). Plots were created in python 3.7 with pandas 0.24.2 using the seaborn 0.9.0 package.

**Protein structural models**. Models of protein complexes with mapped residue pairs (Supplementary Data 8) were prepared with xiVIEW[40], python 3.7 with pandas 0.24.2 and ChimeraX 0.92[41].

All structural PPI models were downloaded from the protein data bank (https://www.rcsb.org/): PDB 5t4O [https://doi.org/10.2210/pdb5T4O/pdb] (ATP synthase[42]), PDB 6RKW [https://doi.org/10.2210/pdb6RKW/pdb] (DNA gyrase[43]), PDB 4PKO [https://doi.org/10.2210/pdb4PKO/pdb] (GroEL[44]), PDB 4S20 [https://doi.org/10.2210/pdb4S20/pdb] (RapA[45]), PDB 6RIN [https://doi.org/10.2210/pdb6RIN/pdb] (GreB[46]), PDB 5MS0 [https://doi.org/10.2210/pdb5MS0/pdb] (NusG[47]), PDB 6FLQ [https://doi.org/10.2210/pdb6FLQ/pdb] (NusA[48]) and PDB 4ZH3 [https://doi.org/10.2210/pdb4ZH3/pdb] (RpoD[49]).

**Database search and FDR calculation for sulfo-SDA crosslinked pulldowns**. A recalibration of the precursor m/z was conducted based on high-confidence linear peptide identifications[37]. The re-calibrated peak lists were searched against the sequences of proteins identified in a given pull-down and with an iBAQ ≥ 5e6 along with their reversed sequences (as decoys) using xiSEARCH (v.1.7.6.2) for identification. MS-cleavability of the sulfo-SDA crosslinker was considered[50]. Final crosslink lists were compiled using the identified candidates filtered to 2% FDR on residue pair-level and 5% on PPI level with xiFDR v.2.1.5[17].

**RNA polymerase binding site of YacL**. An I-TASSER41 (v.5.1)[51] model for YacL was generated with default settings based on the Uniprot sequence (see above). DisVis (v.2.0)[52] ran under default settings, with YacL as scanning model and fixed model PDB 6C6U [https://doi.org/10.2210/pdb6c6u/pdb][53] with residue 118–127 of NusG modelled using the Modeller[54] plug-in in Chimera[55]. Residue pairs of YacL to RNAP and NusG were used as restraints with a minimal distance of 2 Å, and a maximal distance of 30 or 20 Å for DSSO/BS3 and sulfo-SDA, respectively. The density displayed in Fig. 4d corresponds to the accessible interaction space with 14 satisfied restraints. The I-TASSER model was placed for visualisation purposes only.

**Reporting summary**. Further information on research design is available in the Nature Research Reporting Summary linked to this article.

## Data availability

Raw data and MaxQuant outputs from quantitative proteomics SEC-MS experiments were deposited with the ProteomeXchange Consortium partner repository jPOSTrepo under the accession codes JPST000843[56] and PXD019004. Raw data and MaxQuant outputs from quantitative proteomics AP-MS experiments were deposited with the ProteomeXchange Consortium partner repository jPOSTrepo under the accession codes JPST001090[56] and PXD024146. All raw data, peak lists and search result files from BS3/DSSO crosslinking experiments in the SEC fractions and after multidimensional fractionation were deposited with the ProteomeXchange Consortium partner repository jPOSTrepo under the accession codes JPST000845[56] and PXD019120. All raw data, peak lists and search result files from affinity-enrichment and crosslinking experiments were deposited with the ProteomeXchange Consortium partner repository jPOSTrepo under the accession JPST001091[56] and PXD024148. We accessed the STRING database (v10.5) via https://string-db.org/. The new link for this version is https://version-10-5.string-db.org/. The used resource can be downloaded using the following link: https://version-10-5.string-db.org/download/protein.links.detailed.v10.5/511145.protein.links.detailed.v10.5.txt.gz. Models from the protein data bank (PDB) can be found under the following links: PDB 5t4O [https://doi.org/10.2210/pdb5T4O/pdb] (ATP synthase[42]), PDB 6RKW [https://doi.org/10.2210/pdb6RKW/pdb] (DNA gyrase[43]), PDB 4PKO [https://doi.org/10.2210/pdb4PKO/pdb] (GroEL[44]), PDB 4S20 [https://doi.org/10.2210/pdb4S20/pdb] (RapA[45]), PDB 6RIN [https://doi.org/10.2210/pdb6RIN/pdb] (GreB[46]), PDB 5MS0 [https://doi.org/10.2210/pdb5MS0/pdb] (NusG[47]), PDB 6FLQ [https://doi.org/10.2210/pdb6FLQ/pdb] (NusA[48]), PDB 4ZH3 [https://doi.org/10.2210/pdb4ZH3/pdb] (RpoD[49]), PDB 6C6U [https://doi.org/10.2210/pdb6c6u/pdb] (RNAP-NusG[53]). Source data are provided with this paper.

## Code availability

The xiFDR version[57] used in this manuscript (v2.0.dev) is available via Zenodo at https://doi.org/10.5281/zenodo.4682917. More recent xiFDR versions can be downloaded from https://github.com/Rappsilber-Laboratory/xiFDR or https://www.rappsilberlab.org/software/xifdr/.

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

## Acknowledgements

We would like to thank Richard Scheltema, Alexander Leitner, Andrea Sinz, Michael Hoopmann, Marc Wilkins, Fan Liu, Henning Urlaub, James Bruce, Si-Min He, Meng-Qiu Dong and Dermot Harnett for comments on the manuscript. We thank Tabea Schütze for fermenting *E. coli*. The work was funded by the Deutsche Forschungsgemeinschaft (DFG, German Research Foundation) under Germany's Excellence Strategy—EXC 2008—390540038—UniSysCat, by grant no. 392923329/GRK2473, grant no. 426290502 and by the Wellcome Trust through a Senior Research Fellowship to J.R. (103139). The Wellcome Centre for Cell Biology is supported by core funding from the Wellcome Trust (203149).

## Author contributions

F.O., L.S., S.L. and J.R. designed the experiments; L.S., F.W. and F.O. prepared the samples. L.S. did the affinity-enrichment experiments. L.S., S.L. and L.F. collected and

processed Crosslinking MS data; L.F. designed and implemented xiFDR software; S.L., L.S., F.O. and J.R. prepared figures and wrote the manuscript with input from all authors.

## Funding

## Competing interests
The authors declare no competing interests.
