## [Peer Review File · Nature Communications]

REVIEWER COMMENTS

Reviewer #1 (Remarks to the Author):

In their manuscript, Lenz et al. make four contributions:

1) A clever experimental setup relying on SEC fractionation that allows to estimate the FDR of PPIs independently of the common decoy sequences approach.

2) A clear demonstration of the need to separate the FDR estimation of the heteromeric cross-links from that of the self links. I should mention that this understanding has formed in the XL-MS community over the last 4-5 years, and is already being practiced (e.g. Chavez JD, et al., Cell Syst. 6:136-141, 2018 calculate a different FDR for the inter- and intra-links). I appreciate the thorough study of this issue by the authors, but I think it would be fair if the introduction included a more explicit statement that this notion is not new to this work.

3) A clear demonstration of the need to estimate the FDR at the PPI level differently than previous levels. As large interactome studies by XL-MS are emerging, this is an important issue.

4) A large and high-quality PPI network based on XL-MS of an important model organism.

Given these contributions, I think the manuscript is of interest to several sub-fields in biology and merit publication in Nature Communications. I did, however, found several parts to be confusing and would like to make the following suggestions:

Major:

1. The authors are in a unique position where they can compare their FDR estimate of “non-crosslinkable” hits with the “traditional” way of decoy sequences. Yet, such direct comparison is not shown. Figure 2a,b only shows the “non-crosslinkable” estimate (if I understood correctly), and Figure S2 shows the two estimates on two different panels (b and d). I think that a direct comparison of the FDR values of the two approaches should be given on the same graph, preferably in the main

text. I use decoy sequences frequently, and I find it of great interest to see how they perform compared to an independent benchmark (which is unique to this study).

2. I did not understand to what the numbers in Figure 3b refer to. These are percentages of what? Why does the sum of the left Venn diagram not adding to 100%?

3. I understood Figure S2 to be an expansion of Figure 2a,b for different CSM-FDR values other than 5%. If this is correct: Shouldn't the Y-axis label be "Estimated FDR" ? Shouldn't the color legends be reversed ('CSM' in strong red and 'PPI' in pink)? There is inconsistency between values in S2 and 2a,b. For example, the "Residue Pair" value at 5% is 30% in S2a, but this value is not occurring at all in 2a,b. How can that be?

4. I think that a graph showing the dependency of the number of PPIs vs. the FDR cutoff in the high-quality dataset would be illuminating (even in the SI).

Minor:

1. It is very curious that GroS is not found as interactor of GroL (Fig. S7). Do the authors have any theory as to why?

2. Why were AceA and TnaA removed from Figure 3? To my understanding they are enzymes and not chaperones with numerous interactors.

Reviewer #2 (Remarks to the Author):

Review of Lenz et al., "Reliable identification of protein-protein interactions by crosslinking mass spectrometry"

In the study presented here Lenz et al. investigated protein interactions in E. coli lysates by protein crosslinking and mass spectrometry (termed crosslinking MS). The major focus of the work is to

evaluate a reliable false-positive discovery rate (FDR) estimation/calculation in protein crosslinking and mass-spectrometric identification of the crosslinked peptides. In general I appreciate the idea behind this work and the subsequently applied analytical workflow that the authors used to achieve this.

The authors first applied protein size-exclusion chromatography to separate protein complexes in *E. coli* cell lysate, a workflow that is also described as “complexome profiling”. The authors crosslinked the SEC fraction with BS3 or DSSO and purified crosslinked peptide by ion-exchange chromatography. Crosslinked peptides were analysed by LC-MS and data were searched by the group's software. In addition, the authors analysed each SEC fraction by LC-MS without prior crosslinking in order to achieve identification of proteins with their abundance (iBAQ values) in each fraction. The authors hypothesize (in my view, correctly), that only crosslinks between proteins/peptides are valid if they occur among the proteins in each SEC fraction. By this means they are also able to evaluate more precisely crosslinks between two different proteins rather than crosslinks within the same protein, which leads (again, in my view correctly) to a separate inspection of “inter” crosslinks (here termed heteromeric crosslinks) and “intra” crosslinks (here termed self-links). The relation between proteins identified and crosslinks identified in each fraction allows reliable estimation of the FDR for crosslinked peptides of two different proteins in a data-base search, since only those crosslinks should be “real” that belong to those proteins in the respective SEC fractions. The authors' approach also enables them to define not only the FDR of crosslinked peptides but also that of protein–protein interaction, similar to conventional proteomic experiments in which the database search can distinguish between peptide and protein FDRs. The authors demonstrate the suitability of their concept by identifying 590 different protein–protein interactions of an *E. coli* lysate separated by protein size-exclusion chromatography at 1% FDR. Finally, the authors modelled a yet uncharacterised *E. coli* protein YacL in complex with the RNA polymerase and its associated factors.

As stated above, I appreciate the basic concept on this work and I consider that it addresses an issue that is currently under intensive discussion in the field of protein crosslinking. The data are well presented and are convincing. Nonetheless, because the manuscript focusses mainly on generating a reliable FDR for PPIs within a specific software (xiFDR) and does not introduce novel (biochemical or protein chemical) techniques, I wonder whether the message of the manuscript is not perhaps rather restricted and hence more appropriate for a more specialised journal. Regrettably, the novelty of PPIs in *E. coli* is restricted by showing an as yet uncharacterised interaction of YacL with the RNA polymerase based only on crosslinking. The TASSER model generated leaves much room for interpretation and the function of the crosslinked YacL remains completely unclear.

Reviewer #3 (Remarks to the Author):

Swantje Lenz, Ludwig R. Sinn, Francis J. O'Reilly, Lutz Fischer, Fritz Wegner, Juri Rappsilber
Reliable identification of protein-protein interactions by crosslinking mass spectrometry

Submitted to Nature Communications

Review Summary

This manuscript by Lenz et al. proposes a protein fractionation-based experimental method to validate a target-decoy approach (TDA) to estimating the false discovery rates (FDRs) of interacting protein pairs identified by crosslinking mass spectrometry (MS) for complex samples.

This method can be considered as a natural and even perfect extension of the synthetic peptide based method proposed in Beveridge-NC-2020, extending the validation from the peptide pair level to the protein-protein-interaction (PPI) level.

With this experimental method, and another computational trap-database method in the same spirit, this manuscript tries to prove two propositions:

Proposition 1: Neither the naïve/combined FDR control for self and heteromeric crosslinks at crosslink-spectrum match (CSM) or peptide pair levels, nor the heteromeric/ separate FDR control at levels lower than PPI, can reliably estimate the PPI FDR.

Proposition 2: With heteromeric PPI FDR control, xiFDR can reliably estimate the PPI FDR for crosslinking MS.

The authors by and large succeeded in proving both, especially Proposition 1, but they could do better by providing further evidence, e.g. using a second search engine. In particular, the proof of Proposition 2 seems too optimistic and needs discussion for a comprehensive view.

I support its publication in Nature Communications. Detailed comments, including contributions, major concerns, and minor concerns, are as follows.

Contributions

The manuscript advocates two points: (1) separate FDR control for self and heteromeric crosslinks, and (2) multi-level FDR control for CSM, peptide pair (PP), residue pair (RP) and PPI levels. These two points are common sense in routine protein identification, and not novel even in crosslink identification. However, they are still worth emphasizing, since they are yet to be embraced by all software developers or users. As noted in Supp Table 1, in the past two years, although most papers use separate FDR control, several papers still use combined FDR control; few papers use residue pair or PPI FDR control, which are routinely supported only by xiFDR. A few years ago, only a small number of crosslinks can be identified, and hence PPI FDR control is infeasible. But now, substantially more crosslinks can be identified, and hence higher level FDR control should be put on the agenda.

The difficulty lies less in the implementation of the two points into software, since the key formula for crosslink TDA FDR estimation, i.e., $FDR = (\#TD - \#DD)/(\#TT)$, has been proved

independently in Walzthoeni-NM-2012 and Yang-NM-2012, and later in Fischer-AC-2017. When merging CSMs to PPIs, xiFDR only considers protein-unique peptides, and hence the merging is straightforward, and the same TDA FDR formula is extended to all levels.

The difficulty lies more in the validation of the implemented TDA FDR estimation. Among the various attempts, Beveridge-NC-2020 is one of the milestones for experimental validation of TDA FDR estimation at CSM and peptide pair levels, in a way that cleverly avoids the laborious manual annotation of spectra.

Beveridge-NC-2020's validation method is designed as follows: Synthesize 95 peptides of a seed protein, classify them into 12 groups, crosslink among peptides in each group, and then pool the crosslinks together for MS sampling, identify them by searching a database of all peptides of the seed protein together with some more. The principle is simple: crosslinks are generated in a small space (intra-group), but are searched in a large space (inter-group); therefore, any identified peptide pair crosslink is considered plausible if they are in the same peptide group, and considered non-crosslinkable otherwise. This provides an FDR estimation independent of TDA, and hence can serve as a validation of the latter. Beveridge-NC-2020's method is limited by its higher cost of synthetic peptide crosslinking and poor scalability, and hence cannot be used to mimic complex samples.

This manuscript proposes an experimental method that can be considered as a natural and even perfect extension of Beveridge-NC-2020 to complex samples. The design procedure is quite similar: Divide the *E. coli* sample into 44 SEC fractions, crosslink among proteins in each fraction, then pool the fractions together for MS sampling, identify crosslinks by searching the whole database of *E. coli*. Again, crosslinks are generated in a small space (intra-fraction), but are searched in a large space (inter-fraction). Therefore, any identified peptide pair crosslink or the corresponding PPI is considered plausible if the crosslink related protein pair is in the same fraction, and considered non-crosslinkable otherwise. This provides an FDR estimation independent of TDA, and hence can serve as a validation of the latter. In particular, this method can be applied to complex samples.

The method is applied to validating the two propositions with xiSearch + xiFDR, with largely successful results:

For Proposition 1: If 5% TDA FDR is controlled at levels lower than PPI, either separating or combining self and heteromeric crosslinks, then the estimated PPI FDR by the new experimental method will be at least over 10% (heteromeric RP) and as large as over 35% (naïve CSM), significantly higher than the target 5% PPI FDR.

For Proposition 2: If 5% TDA FDR is controlled at the PPI level, then the estimated PPI FDR by the new experimental method will be 6.6% (BS3) or 4.9% (DSSO), only a slight deviation from the target 5% PPI FDR.

This manuscript also proposes a trap-database method, another validation method in the same spirit of Beveridge-NC-2020 and the experimental method of this paper: generate crosslinks in a small space (the original database) and search crosslinks in a large space (the original database plus a trap database); crosslinks within the small space are plausible, and those outside are non-crosslinkable. Therefore, this trap-database method is also independent of TDA and can be

used to validate the TDA FDR, and the validation results are similar to those of the experimental method as listed above. In particular, the trap-database method is purely computational, and hence can be more widely used than the protein fractionation based experimental method, since the latter can only be used with its specific data set, while the former can be used with arbitrary data sets. However, the trap-database method can be considered as having only one fraction, and hence should be weaker than the multi-fraction based experimental method in terms of discriminating power, which is a conjecture to be proved.

In short, this manuscript contributes two TDA-independent validation methods, one is experimental, and the other is computational, both of which can be used to estimate crosslink FDR at all levels and for complex samples. The two methods share the same design principle, i.e. to generate in a small space and to validate in a large space. There are other validation methods for complex samples, e.g. the ¹⁵N-labeling method in Chen-NC-2019, the multi-fragmentation method in Zhao-JPR-2020, which share a different principle: to provide more positive control information for the target, rather than to provide more negative control information as the two methods proposed in this paper. All these methods are shaping the crosslinking MS towards the right direction.

Major Concerns

There are five occurrences of “reliable” or “reliably” in this manuscript:

- Title: “**Reliable** identification of protein-protein interactions by crosslinking mass spectrometry.”
- Abstract: “Using a carefully controlled large-scale analysis of *Escherichia coli* cell lysate, we demonstrate that false-discovery rates (FDR) for PPIs identified by crosslinking mass spectrometry can be estimated **reliably**.”
- Results: “In contrast, first merging CSMs for each PPI and then assessing the FDR gave more **reliable** results: 6.6% and 4.9% false PPIs when applying 5% PPI-FDR (Fig. 2e) for BS3 and DSSO, respectively.”
- Discussion: “In this work, we experimentally demonstrated that Crosslinking MS can **reliably** identify PPIs.”
- Discussion: “Crosslinking MS for mapping PPIs now has a **reliable** FDR estimation procedure.”

Question: How **reliable** is the TDA FDR control of xiFDR?

Does it mean that the PPI FDR estimated by xiFDR will deviate from the true value by no more than 2% absolutely? Does it mean that the PPI FDR estimated by the protein fractionation based experimental method is **reliable**?

To trace back, how **reliable** are TDA FDR controls in *routine protein identification*?

I do not think that TDA FDR controls in routine protein identification are reliable:

- (1) The TDA FDR model can only model random matching errors, but cannot model homologous matching errors. That is, the model contains a systematic error.
- (2) The TDA model has a so-called 1:1 hypothesis, i.e. a random matching will match targets and decoys with the same probability. However, there are several ways to construct decoys, e.g., reversing, shuffling, or reversing with local swapping (“the enzyme specific amino acids were swapped with their preceding amino acid” as used in MaxQuant and this manuscript). Nobody knows whether the 1:1 hypothesis will be satisfied in practice, given a target sequence database coupled with a decoy construction method, given a mass spectrometer and a search engine.
- (3) Even if all spectra are matched to the correct peptides, further mapping peptides to proteins is not simple. Shared peptides will result in different ways of protein grouping. Very very large data sets will produce an anomaly under the classic TDA FDR model: more data result in less results.

Although I do not trust TDA FDR methods, I support their use since they are simple and universal, and in particular, *no other methods are more reliable and meantime as simple and universal*. However, currently computational proteomics bases its precision solely on the TDA-FDR model, which is quite dangerous to me.

Crosslinking MS is more complex than routine protein identification, and hence its TDA-FDR control cannot be simpler, and should be used with greater caution. While I fully support Proposition 1, since it has strong evidence both inside and outside this manuscript, I am deeply concerned that both the authors and the readers may take a simplistic view of Proposition 2, for which the evidence is not strong enough and may never be. I suggest the authors gather new evidence to achieve a comprehensive understanding of Proposition 2, and discuss limitations of the two methods. Even if the new evidence does not support Proposition 2 as the available evidence in this manuscript did, I still support the publication of this manuscript and the adoption of xiFDR.

Suggestion 1. Use at least another crosslink search engine to test the two propositions and the two validation methods proposed in this manuscript.

The two validation methods are universal, similar to the method in Beveridge-NC-2020. However, Beveridge-NC-2020 tested its method with five search engines including xiSearch, Stavrox/Merox, XlinkX, pLink, Kojak and two variants for Kojak and for Merox. In comparison, this manuscript only tested one search engine xiSearch. This manuscript claims that xiFDR v2.0 is crosslink search software independent, which needs direct evidence.

Suggestion 2. Besides PPI FDR, validate FDRs at lower levels including CSM, PP and RP, and besides FDR for heteromeric crosslinks, validate FDR for self crosslinks.

FDRs at any information level (CSM, PP, RP and PPI) or type (self, heteromeric) may be useful and necessary. However, in this manuscript, only PPI FDR of xiFDR is validated by the two validation methods, although it seems that the same methods can be used to validate TDA FDRs at all other levels. Now that this manuscript claims that the PPI FDR can be estimated reliably, I think most readers may take it for granted that FDRs at all other levels can be estimated reliably,

too. Please prove or disprove this impression.

Suggestion 3. Try another way to estimate the FDRs by the two validation methods with the *E. coli* data set.

$$\begin{aligned} Error_{PPI,non-crosslinkable} &= \frac{n_{false}}{n_{plausible} + n_{false}} \cdot \frac{KR_{plausible} + KR_{false}}{KR_{false}} \\ &= \frac{n_{false}}{n_{plausible} + n_{false}} \cdot 1.09 \end{aligned}$$

$$\begin{aligned} Error_{PPI,entrapment} &= \frac{n_{entrapment}}{n_{E.coli} + n_{entrapment}} \cdot \frac{KR_{E.coli} + KR_{entrapment}}{KR_{entrapment}} \\ &= \frac{n_{entrapment}}{n_{E.coli} + n_{entrapment}} \cdot 1.998 \end{aligned}$$

Let's discuss a general framework for error estimation by a validation method. Please refer to (1) Chen-NC-2019 (<https://doi.org/10.1038/s41467-019-11337-z>), Supplementary Note 3, Deducing an NaN-FDR for a search engine independent of the TDA-FDR. (2) Zhou-JPR-2019 (<https://doi.org/10.1021/acs.jproteome.8b00993>).

Given a set of N identifications of any kind, with T correct and F incorrect. Hence $N = T + F$, and the error rate is F/N .

Given a validation method with a false positive rate e_1 and a false negative rate e_2 for the data set. The validation method detects S suspects from the N identifications; specifically, it detects $T \cdot e_1$ false alarms and $F \cdot (1 - e_2)$ true alarms. Hence $S = T \cdot e_1 + F \cdot (1 - e_2)$.

By simple deduction, $F/N = (S/N - e_1) / (1 - e_1 - e_2)$.

The ratio F/N is the error rate to be estimated, denoted by $Error_{estimated}$. And the ratio S/N is the detected error rate, denoted by $Error_{detected}$. Then

$$Error_{estimated} = (Error_{detected} - e_1) / (1 - e_1 - e_2).$$

For the validation method in Beveridge-NC-2020 and the two validation methods in this manuscript, they are all based on the entrapment validation approach, and it can be reasonably assumed that $e_1 \approx 0$, while e_2 may be quite different for each method. Then

$$Error_{estimated} = Error_{detected} / (1 - e_2).$$

Compare the above general formula with the specific formula of the experimental validation method:

$$Error_{PPI,non-crosslinkable} = Error_{estimated},$$

$$\frac{n_{false}}{n_{plausible} + n_{false}} = Error_{detected},$$

$$\frac{KR_{plausible} + KR_{false}}{KR_{false}} = 1.09 = (1 - e_2)^{-1}.$$

The formula of the trap-database method can be interpreted similarly.

Relatively speaking, $Error_{detected}$ can be calculated objectively, and whether $Error_{estimated}$ is reliable depends on whether e_2 is estimated reliably.

This is why I trust Proposition 1 more than Proposition 2, since Proposition 1 can still be proved with no need of estimating e_2 . Specifically, naïve or heteromeric 5% FDR at lower levels will still result in $Error_{detected} \gg 5\%$ at the PPI level. No matter how e_2 is estimated, $e_2 \geq 0$, and hence $Error_{estimated} \geq Error_{detected}$. Therefore, $Error_{estimated} \geq Error_{detected} \gg 5\%$.

In contrast, Proposition 2 may no longer be true if the estimation of e_2 (the equivalent of 1.09 or 1.998) is not reliable. As stated in the manuscript, 11% of the proteins found in 590 PPIs at 1% PPI FDR had no self-links, which is a sign to me that the actual FDR may be higher than the expected 1%. Although 1.09 and 1.998 are reasonable and clever estimations, I think it might be more convincing if another independent estimation of e_2 can be tried and hence another independent estimation of FDR can be obtained.

The pValid paper Zhou-JPR-2020 might be a reference. Shift the precursor masses by some Daltons, then try the same search process as before. All search results shall be incorrect, among which some are plausible, and the other are non-crosslinkable. You can estimate e_2 at every level from CSM to PPI, and e_2 may have a different value at different level, rather than the rigid 1.09 or 1.998.

For the trap-database method, actually you can use arbitrary databases as entrapment, which is no longer required to be similar to the original database, as long as you can estimate the corresponding e_2 .

There is no free lunch. For Beveridge-NC-2020, it seems that $e_2 = 0$, and the error rate can be estimated more objectively and straightforwardly ($Error_{estimated} = Error_{detected}$), but the method cannot be extended to large scale or used for complex samples. For the two methods proposed in this manuscript, they are applicable to complex samples, but they have to estimate e_2 , a key parameter of the validation method.

Anyway, large-scale benchmark data sets or validation methods are difficult to design, yet they are indispensable to precision proteomics, including precision crosslinking proteomics. This manuscript made a solid step forward in the right direction. While the TDA FDR model will accompany our journey, it is only a convenient walking stick, and cannot be solely relied upon.

Minor Concerns

➤ General:

- (1) The original version of this manuscript, posted in bioRxiv, was apparently prepared as Brief Communications, and hence the statements and the references were concise. But now, the manuscript was prepared as Article, the statements in the text and figure legends and the references should be enriched accordingly.
- (2) Please add numbers for all bars in all bar chart displays, including Fig. 1b, Fig. 2,

Supp Figs. 1b and 1c (the two dotted lines), Supp Figs. 2, 3, 4. Additional tables may be used when necessary.

- **“This target-decoy approach has been adapted for Crosslinking MS¹⁴⁻¹⁷.”**

Yang-NM-2012 (<https://www.nature.com/articles/nmeth.2099>) should be cited. See its Abstract (“**pLink reliably estimates false discovery rate in cross-link identification...**”), its text (“**The FDR calculation is based on in silico cross-linking of forward (F) and reversed (R) peptide sequences, computed as the number of identified F-R and R-F cross-links subtracted by the number of identified R-R cross-links, then divided by the number of identified F-F cross-links (details in Supplementary Note).**”), and its Supplementary Information (Supplementary Figure 15 “FDR estimation” and Supplementary Note “Estimation and control of FDR” at page 43).

- **“The first, whether to consider crosslinks between peptides within one protein sequence (self-links, including homomeric crosslinks) separately from crosslinks between protein sequences (heteromeric crosslink)¹⁵.”**

Beside your reference 15, your reference 4 or Chen-NC-2019 should also be considered as reference, which conducted both theoretical and experimental studies concerning separate control, as shown explicitly in its text and supplementary information: “**Several studies have shown that the use of separate control of the FDR for inter-protein and intraprotein identifications, rather than global control, is an effective means of improving credibility of inter-protein results^{11,12}. Our data analysis confirmed this and found that intra-protein PSMs and inter-protein PSMs show different changes when switching from global FDR control to separate FDR control (Fig. 3d, e). For intra-protein PSMs, more results were reported under separate FDR control and its percentage of NaN ratios was only slightly higher than that under global FDR control (Fig. 3d). For interprotein PSMs, many fewer results were reported under separate FDR control and its percentage of NaN ratios decreased notably (Fig. 3e). This phenomenon also existed in the results of Protein Prospector (Supplementary Fig. 8), which recommended separate FDR control in its study¹². Furthermore, our theoretical analysis about the relationship between the global FDR and subgroup FDRs of intra-protein and inter-protein identifications was accordant with the above experimental phenomenon; please see Supplementary Note 2 for details.**” The reference 11 therein is your reference 15, and reference 12 therein or Trnka-MCP-2014 should also be considered as your reference.

- **“In our data, the chance of matching a decoy crosslink (random) within the heteromeric crosslinks is 10.6 times higher than within the self-links (Fig. 1b). Controlling FDR in the total set of CSMs, and then selecting only heteromeric matches thus enriches for false positives. This leads to a large underestimation of the error within heteromeric CSMs, which describe PPIs (Fig. S1). Consequently, heteromeric crosslinks must be considered separately from self-links during FDR estimation.”**

First, under what condition was 10.6 obtained? There was no explanation, either in the text or in the figure legend.

Second, do you mean that the statement of the first sentence is a sufficient condition for the statement of the second sentence, and further for the third sentence? Suppose that there were ten times fewer self-links in the sample, did your statements and deductions still hold true? Please refer to “Supplementary Note 2. The relationship between the global FDR and the subgroup FDRs of intra-protein and inter-protein cross-linked identifications” of Chen-NC-2019 (your reference 4).

➤ **Of note, in our experimental control 87% of false PPIs involved proteins that were seen only with heteromeric crosslinks, i.e. that lacked self-links (Fig. 2c). In the entrapment control this number increased to 100% (Fig. 2c).**

How were the false PPIs obtained? Were they obtained by first filtering CSMs at a naïve 5% CSM-FDR, then merging the heteromeric CSMs into PPIs, and finally classifying the PPIs into false (non-crosslinkable) and true (plausible)?

➤ **If observed at all, ‘heteromeric only’ proteins had a lower median abundance than all proteins in the sample suggesting that they are enriched in random matches (Fig. 2c).**

Error: Fig. 2c should be Fig. 2d.

Again, how did you obtain the “‘heteromeric only’ proteins” and “all proteins”? At naïve CSM-FDR or at PPI-FDR? 1% or 5%? With the *E. coli* proteome database of 4350 or 1926 proteins? With additional pruning of non-crosslinkable PPIs?

In particular, how did you obtain the self-only proteins? These proteins should only come from self crosslink CSMs, but you did not clearly define how to estimate and control the FDRs for self crosslinks at various levels, from the CSM level to the protein levels.

➤ **As predicted, CSMs rarely corroborated each other in false PPIs while plausible PPIs were supported by multiple CSMs (average 1.2 versus 4.6 CSMs), irrespective of the crosslinker (Fig. 2a,b).**

For an Article in Nature Comm, there is enough space for complete description. Hence I suggest that besides describing 1.2 and 4.6 for BS3, 1.3 and 5.2 for DSSO should also be added. Furthermore, the explanations in the text should be added to the legends to Fig. 2a and 2b.

➤ **In contrast, first merging CSMs for each PPI and then assessing the FDR gave more reliable results: 6.6% and 4.9% false PPIs when applying 5% PPI-FDR (Fig. 2e) for BS3 and DSSO, respectively. This also applies to other FDR thresholds and the entrapment control which indicated an error of 1.4% for BS3 and 3.6% for DSSO when applying 5% PPI-FDR (Fig. S2).**

If no iBAQ control is applied for non-crosslinkable PPI judgement, what are the numbers in

Fig. 2e?

There is no such error information as 1.4% or 3.6% in Fig. S2.

➤ In previous studies, the quality of identified crosslinks was assessed by measuring interresidue distances in known protein structures. **However, for proteome-wide crosslinking studies, this approach is inherently biased towards true interactions as 'real' crosslinked matches are likely to be enriched in these known complexes.** The majority of random PPIs are neglected by this FDR evaluation method, making this approach completely inadequate for reliable PPI error estimation.

The sentence in the middle is hard to understand.

➤ **Figure 1: Considerations for crosslinked PPI-FDR and experimental workflow. b) Fraction of decoys in 10 random picks of 100 self and 100 heteromeric PPIs from the search output before FDR filtering.**

Please complement the bars with numbers, and describe more exactly how to obtain the results, e.g. before FDR filtering at the PPI level yet after FDR filtering at the RP level 5%? Or before no FDR filtering at any level and after merging all top-1 CMSs into PPIs?

➤ **Figure 2: Comparative analysis of different methods of FDR estimation in Crosslinking MS.**

Please complement all bars with the corresponding numbers.

Please explain all the numbers in the figure, such as 5.2 and 1.3 in a), 4.6 and 1.2 in b), 339 and the other 5 numbers in d).

Please explain the inconsistency between the numbers in Fig. 2d and Supp Fig. S4C.

“Average of BS3 and DSSO data (separated in Fig. S4c)”: “Fig. S4c” should be “Fig. S3, a and b.”

➤ **LS-MS protein ID: ACG target set to $1 \cdot 10^5$.**

➤ **LC-MS for crosslink ID: ACG of $5 \cdot 10^4$**

ACG should be AGC.

➤ **Crosslink database search**

“Matches were not filtered for having DSSO-specific signature peaks.” What does this mean? Where are the differences between the algorithms of xiSearch-DSSO and xiSearch-BS3?

“Crosslink sites for both reagents were allowed for side chains of Lys, Tyr, Ser, Thr and the protein N-terminus.” What is the proportion of Lys-Lys crosslinks in self crosslinks and in heteromeric crosslinks at each level of separate FDR control at 5%?

How many MS/MS spectra are sampled for crosslink identification? What is the MS/MS identification rate with xiSearch+xiFDR on your data set, at 5% CSM-FDR separate for self links and heteromeric links?

Are the contamination proteins taken into database search?

“The same database as for the non-crosslinked samples was used. For the entrapment database control the database was extended by the same number (4,353) of human proteins.” The number 4353 was inconsistent with the number 4350 used before in “Quantitative proteomics database search”.

“Decoys were generated for all searches, including the entrapment database. For this, protein sequences were reversed and for each decoy protein the enzyme specific amino acids were swapped with their preceding amino acid.²⁵” Does this simple swapping operation result in notable effect on the identification?

➤ FDR calculation

“Results were filtered prior to FDR to crosslinked peptide matches having a minimum of three matched fragments per peptide, a delta score of 15% of the match score and a peptide length of at least six amino acids.” In your own paper or your reference 6, your criteria are: “Cross-links between two different proteins were analyzed with the following parameters: prefilter cross-links only, Δ score 0.5, minimum number of fragments per peptide five, with eight amino acids as minimum peptide length.” Why do you select different criteria for two papers, according to what principles?

“Additionally, identifications ambiguously matching to two proteins or more were removed.” Among the peptides pairs at 5% peptide pair FDR, what is the proportion of the ambiguous?

“FDR was calculated based on decoy matches by xiFDR (version 2.0dev) using the formula¹⁶: $FDR = (TD - DD) / TT$. Depending on the experiment, FDR was employed on different result levels (CSM, peptide pair, residue pair or protein pair) with defined thresholds.” In routine protein identification, the TDA model has *the 1:1 hypothesis* for random matches in T and D; based on that, in crosslink identification, the TDA model has two hypotheses for random matches, the first is *the 1:2:1 hypothesis* for the random-random crosslinks in TT, TD and DD, and the second is the 1:1 hypothesis for the random-correct crosslinks in TT and TD; based on the two hypotheses, the crosslink TDA formula $FDR = (TD - DD) / TT$ can be deduced. There was one validation experiment for the 1:2:1 hypothesis at CSM level in Yang-NM-2012, Supp Fig 15B. Could you provide a similar validation experiment for the 1:2:1 hypothesis at other levels higher than CSM, especially at the PPI level? For example, delete the 1926 proteins from the 4350 proteins, and use the remaining proteins to search the *E. coli* data set, merge all top-1 crosslink CSMs to higher levels. Your human entrapment database can also be used directly for

such a validation.

“Depending on the experiment, FDR was employed on different result levels (CSM, peptide pair, residue pair or protein pair) with defined thresholds.” For the CSM level, are the CSMs allowed to be redundant or required to be unique? There are two cases of “unique CSMs” occurred in this manuscript, and hence maybe you allow both types. For some search engines, “CSMs” are redundant by default, unless “unique CSMs” are used. There is also one case of “unique residue pairs” in this manuscript.

“For the final PPI network, BS3 and DSSO PPIs were separately filtered to 1% heteromeric PPI-FDR. As the scores cutoff differed between the two datasets, the score was normalised to range between 0 and 1 for subsequent joining of the two lists. The combined list was then filtered again to 1% FDR.” Please describe how to do normalization, joining and further FDR control.

- **Non-crosslinkable control: The question now is what abundance is sufficient. As a heuristic, we removed the lower 5% iBAQ values (iBAQ of 4.3E6). This removed very few (169, 7%) of our identified protein pairs (n.b. at a very loose FDR threshold), i.e. did not change much the outcome of our identification data by generating false negatives. // For any identified protein pair to be considered possible, both proteins had to be found in at least one SEC fraction together with individual iBAQ values above our iBAQ threshold (iBAQ of 4.3E6). Else, they were defined as non-crosslinkable. 544,274 (6% of all theoretically possible PPIs in the E. coli proteome) PPIs are defined as plausible (see Table S4), while 8,914,801 (94%) PPIs are non-crosslinkable.**

First, contrast this with Supplementary Fig. 1: *“Non-crosslinkable control definition. b) Abundance distribution of proteins identified with heteromeric crosslinks at a generous 10% heteromeric peptide pair FDR (best lower iBAQ, see Methods). Protein pairs are accepted as plausible if both proteins reach the 5-percentile in the same fraction, otherwise the pair is defined as ‘non-crosslinkable’.”* Is the iBAQ threshold fixed or dynamic? The threshold is fixed at 4.3E6 across all SEC fractions, or dynamic at 5% in each SEC fraction?

Second, if no such iBAQ control is applied, what will Fig. 2e will be?

Third, I suggest that you use “in the E. coli proteome **of 4350 proteins**” to provide more information for readers to verify your calculated ratios 6% and 94%.

- **Supplementary Figure 1: Non-crosslinkable control definition. a) Illustration of the distribution of decoy matches for self and heteromeric crosslinks. If considered together for FDR calculation, heteromeric decoy matches will be matched more frequently and make up most of the summed decoy matches. If subsequently heteromeric matches are evaluated separately (e.g. for reporting PPIs), their error will be larger than the previously calculated FDR (which would only be correct for the data as a whole). b) Abundance distribution of proteins identified with heteromeric crosslinks at a generous 10% heteromeric peptide pair**

FDR (best lower iBAQ, see Methods). Protein pairs are accepted as plausible if both proteins reach the 5-percentile in the same fraction, otherwise the pair is defined as ‘non-crosslinkable’. 544,274 (6% of all possible PPIs in the E. coli proteome) PPIs are defined as plausible (see Table S4), while 8,914,801 (94%) PPIs are non-crosslinkable.

First, in a), in the FDR filtered results, the numbers of self targets and heteromeric targets seem to be equal, but in practice the number of self targets should be significantly larger than that of heteromeric targets. Why not use numbers of a real instance in your paper for such a demonstration?

Second, do you mean that “If considered together for FDR calculation, heteromeric decoy matches will be matched more frequently and make up most of the summed decoy matches.” will be sufficient to guarantee “If subsequently heteromeric matches are evaluated separately (e.g. for reporting PPIs), their error will be larger than the previously calculated FDR (which would only be correct for the data as a whole).” ?

Third, I suggest that you use “in the E. coli proteome **of 4350 proteins**” to provide more information for readers to verify your calculated ratios 6% and 94%.

➤ **Supplementary Figures 2 and 3**

Please provide additional tables showing the actual numbers of all data points. You have pointed out that “at low FDR thresholds the calculated error is less reliable due to small numbers of false matches,” and hence besides showing the FDR value, also indicate the number of decoys and/or the number of targets in the calculation of FDR. What is the minimal number of PPIs for a reasonable estimation of PPI-FDR?

➤ **Supplementary Figure 4: Properties of final crosslink PPI network.**

Please complement the bar chart display with numbers, for readers’ convenience and for easy correspondence between text and figures.

Fig. S4c seems to be inconsistent with Fig. 2d: (1) In Fig. S4c, self-only crosslinked proteins had the lowest median abundance, while in Fig. 2d, heteromeric-only crosslinked proteins had the lowest median abundance ; (2) The numbers at the top of Fig. S4c, i.e. 644, 274, 34, do not match those of Fig. 2d.

Reviewer: He, Si-Min, smhe@ict.ac.cn, Sept 29, 2020

Reviewer #1

In their manuscript, Lenz et al. make four contributions:

1) A clever experimental setup relying on SEC fractionation that allows to estimate the FDR of PPIs independently of the common decoy sequences approach.

2) A clear demonstration of the need to separate the FDR estimation of the heteromeric cross-links from that of the self links. I should mention that this understanding has formed in the XL-MS community over the last 4-5 years, and is already being practiced (e.g. Chavez JD, et al., Cell Syst. 6:136-141, 2018 calculate a different FDR for the inter- and intra-links). I appreciate the thorough study of this issue by the authors, but I think it would be fair if the introduction included a more explicit statement that this notion is not new to this work.

We thank the reviewer for pointing this out. We have added an additional citation to the introductory sentence "The first, whether to consider crosslinks between peptides within one protein sequence (self-links, including homomeric crosslinks) separately from crosslinks between distinct protein sequences (heteromeric crosslink)^{4,15}."

Note that the use of splitting self-links and heteromeric crosslinks has been used in several publications by a number of labs and we have made an effort to comprehensively list and reference these in Table S1.

3) A clear demonstration of the need to estimate the FDR at the PPI level differently than previous levels. As large interactome studies by XL-MS are emerging, this is an important issue.

4) A large and high-quality PPI network based on XL-MS of an important model organism.

Given these contributions, I think the manuscript is of interest to several sub-fields in biology and merit publication in Nature Communications. I did, however, find several parts to be confusing and would like to make the following suggestions:

Major:

1. The authors are in a unique position where they can compare their FDR estimate of "non-crosslinkable" hits with the "traditional" way of decoy sequences. Yet, such direct comparison is not shown. Figure 2a,b only shows the "non-crosslinkable" estimate (if I understood correctly), and Figure S2 shows the two estimates on two different panels (b and d). I think that a direct comparison of the FDR values of the two approaches should be given on the same graph, preferably in the main text. I use decoy sequences frequently, and I find it of great interest to see how they perform compared to an independent benchmark (which is unique to this study).

We would like to emphasise that our entire manuscript and thus all our figures (including Figure 2a,b) use decoy-based FDRs as their basis. To assess the success of the different ways of computing decoy-based FDRs we cannot rely on decoys, however. So, we used a number of different ways to model false target hits. These are "non-crosslinkable", entrapment, wrong precursor and wrong crosslinker. All four agree in the success or failure of the respective decoy-based FDR methods (see Fig. S2).

2. I did not understand to what the numbers in Figure 3b refer to. These are percentages of what? Why does the sum of the left Venn diagram not adding to 100%?

This has been edited to increase clarity.

3. I understood Figure S2 to be an expansion of Figure 2a,b for different CSM-FDR values other than 5%. If this is correct: Shouldn't the Y-axis label be "Estimated FDR" ?

Figure S2 is actually an extension of Figure 2e. We have now simplified Figure S2. The y-axis represents the error on PPIs, calculated with the respective control, while the information level, for which 5% decoy-based FDR was applied, is shown on the x-axis.

Shouldn't the color legends be reversed ('CSM' in strong red and 'PPI' in pink)? There is inconsistency between values in S2 and 2a,b. For example, the "Residue Pair" value at 5% is 30% in S2a, but this value is not occurring at all in 2a,b. How can that be?

Figure 2a,b show the error of a 5% naive CSM FDR on different result levels, while Figure S2 only represents the error on PPI level for different FDR methods. Therefore, the 36% error on PPIs in 2a corresponds to naive, non-redundant CSM-FDR in Figure S2a (the non-crosslinkable datapoint).

4. I think that a graph showing the dependency of the number of PPIs vs. the FDR cutoff in the high-quality dataset would be illuminating (even in the SI).

We agree, please see Figure S4a.

Minor:

1. It is very curious that GroS is not found as interactor of GroL (Fig. S7). Do the authors have any theory as to why?

GroEL and GroES probably separate from each other in our fractions. This makes sense from our molecular understanding of the complex where GroEL first has cooperative binding of ATP to the seven subunits of its ring and is then primed for GroES binding (PMID: 26422689). Our experiment is totally depleted for ATP (after separation by SEC) so it is plausible that this ATP-dependent interaction is lost in the fractions due to lack of ATP.

2. Why were AceA and TnaA removed from Figure 3? To my understanding they are enzymes and not chaperones with numerous interactors.

AceA and TnaA were not visualised in Figure 3a for clarity. Both proteins were seen with a large number of crosslink partners and cluttered the network, obstructing view from other PPIs. In addition, we suspect many of the links involving these two proteins to be artefacts as these were the most abundant proteins in their fractions (and also the most abundant proteins in the entire dataset) and ran in SEC at roughly the molecular mass of single proteins (later than fraction 40). Their crosslinks are listed as part of supplementary table S5.

Reviewer #2

Review of Lenz et al., "Reliable identification of protein-protein interactions by crosslinking mass spectrometry"

In the study presented here Lenz et al. investigated protein interactions in E. coli lysates by protein crosslinking and mass spectrometry (termed crosslinking MS). The major focus of the work is to evaluate a reliable false-positive discovery rate (FDR) estimation/calculation in protein crosslinking and mass-spectrometric identification of the crosslinked peptides. In general I appreciate the idea behind this work and the subsequently applied analytical workflow that the authors used to achieve this.

The authors first applied protein size-exclusion chromatography to separate protein complexes in E. coli cell lysate, a workflow that is also described as "complexome profiling". The authors crosslinked the SEC fraction with BS3 or DSSO and purified crosslinked peptide by ion-exchange chromatography. Crosslinked peptides were analysed by LC-MS and data were searched by the group's software. In addition, the authors analysed each SEC fraction by LC-MS without prior crosslinking in order to achieve identification of proteins with their abundance (iBAQ values) in each fraction. The authors hypothesize (in my view, correctly), that only crosslinks between proteins/peptides are valid if they occur among the proteins in each SEC fraction. By this means they are also able to evaluate more precisely crosslinks between two different proteins rather than crosslinks within the same protein, which leads (again, in my view correctly) to a separate inspection of "inter" crosslinks (here termed heteromeric crosslinks) and "intra" crosslinks (here termed self-links). The relation between proteins identified and crosslinks identified in each fraction allows reliable estimation of the FDR for crosslinked peptides of two different proteins in a data-base search, since only those crosslinks should be "real" that belong to those proteins in the respective SEC fractions. The authors' approach also enables them to define not only the FDR of crosslinked peptides but also that of protein-protein interaction, similar to conventional proteomic experiments in which the database search can distinguish between peptide and protein FDRs. The authors demonstrate the suitability of their concept by identifying 590 different protein-protein interactions of an E. coli lysate separated by protein size-exclusion chromatography at 1% FDR. Finally, the authors modelled a yet uncharacterised E. coli protein YacL in complex with the RNA polymerase and its associated factors.

As stated above, I appreciate the basic concept of this work and I consider that it addresses an issue that is currently under intensive discussion in the field of protein crosslinking. The data are well presented and are convincing. Nonetheless, because the manuscript focuses mainly on generating a reliable FDR for PPIs within a specific software (xiFDR) and does not introduce novel (biochemical or protein chemical) techniques, I wonder whether the message of the manuscript is not perhaps rather restricted and hence more appropriate for a more specialised journal. Regrettably, the novelty of PPIs in E. coli is restricted by showing an as yet uncharacterised interaction of YacL with the RNA polymerase based only on crosslinking. The TASSER model generated leaves much room for interpretation and the function of the crosslinked YacL remains completely unclear.

We thank the reviewer for their appreciation of the importance of addressing the issue of the correct method for estimation of FDR in crosslinking MS and in particular the application of PPI-FDR in protein interaction screens, which is the central message of our manuscript. For convenience of the field and to facilitate application of our findings we also offer our software implementation, xiFDR.

In addition, we show that our high-fidelity PPI network contains some uncharacterised interactions and highlight the case of YacL to demonstrate the usefulness of the crosslinks as distance restraints to localise the binding interfaces. While the function of YacL is beyond the scope of this manuscript we appreciate that it is desirable to validate this interaction by a second approach. We acquired three strains of *E. coli*, each with a gene tagged with a SPA tag (endogenous locus, leading to a C-terminal tag): rpoB, nusG and yacL. We demonstrate that in affinity purification experiments, tagged YacL leads to the co-isolation of the RNAP and conversely the RNAP subunit, RpoB, and the RNAP binder, NusG, leads to the co-isolation of

YacL (Fig. S10). These are strong independent evidence (also found by Butland *et al.* 2005 (PMID: 15690043)) that YacL is a binder of RNAP. To further localise the binding site of YacL on the RNAP we performed 'high-density' photo-crosslinking on the affinity enriched proteins. The crosslinks from this independent experiment, using affinity enrichment instead of SEC and a different crosslinker, independently validated and further defined our previously observed binding interface (Fig. 4) and thus provide significant lead for further functional studies of the YacL-RNAP interaction.

Reviewer #3

Review Summary

This manuscript by Lenz et al. proposes a protein fractionation-based experimental method to validate a target-decoy approach (TDA) to estimating the false discovery rates (FDRs) of interacting protein pairs identified by crosslinking mass spectrometry (MS) for complex samples. This method can be considered as a natural and even perfect extension of the synthetic peptide based method proposed in Beveridge-NC-2020, extending the validation from the peptide pair level to the protein-protein-interaction (PPI) level. With this experimental method, and another computational trap-database method in the same spirit, this manuscript tries to prove two propositions:

Proposition 1: Neither the naïve/combined FDR control for self and heteromeric crosslinks at crosslink-spectrum match (CSM) or peptide pair levels, nor the heteromeric/ separate FDR control at levels lower than PPI, can reliably estimate the PPI FDR.

Proposition 2: With heteromeric PPI FDR control, xiFDR can reliably estimate the PPI FDR for crosslinking MS.

The authors by and large succeeded in proving both, especially Proposition 1, but they could do better by providing further evidence, e.g. using a second search engine. In particular, the proof of Proposition 2 seems too optimistic and needs discussion for a comprehensive view.

I support its publication in Nature Communications. Detailed comments, including contributions, major concerns, and minor concerns, are as follows.

Contributions

The manuscript advocates two points: (1) separate FDR control for self and heteromeric crosslinks, and (2) multi-level FDR control for CSM, peptide pair (PP), residue pair (RP) and PPI levels. These two points are common sense in routine protein identification, and not novel even in crosslink identification. However, they are still worth emphasizing, since they are yet to be embraced by all software developers or users. As noted in Supp Table 1, in the past two years, although most papers use separate FDR control, several papers still use combined FDR control; few papers use residue pair or PPI FDR control, which are routinely supported only by xiFDR. A few years ago, only a small number of crosslinks can be identified, and hence PPI FDR control is infeasible. But now, substantially more crosslinks can be identified, and hence higher level FDR control should be put on the agenda.

The difficulty lies less in the implementation of the two points into software, since the key formula for crosslink TDA FDR estimation, i.e., $FDR = (\#TD - \#DD)/(\#TT)$, has been proved independently in Walzthoeni-NM-2012 and Yang-NM-2012, and later in Fischer-AC-2017. When merging CSMs to PPIs, xiFDR only considers protein-unique peptides, and hence the merging is straightforward, and the same TDA FDR formula is extended to all levels.

The difficulty lies more in the validation of the implemented TDA FDR estimation. Among the various attempts, Beveridge-NC-2020 is one of the milestones for experimental validation of TDA FDR estimation at CSM and peptide pair levels, in a way that cleverly avoids the laborious manual annotation of spectra.

Beveridge-NC-2020's validation method is designed as follows: Synthesize 95 peptides of a seed protein, classify them into 12 groups, crosslink among peptides in each group, and then pool the crosslinks together for MS sampling, identify them by searching a database of all peptides of the seed protein together with some more. The principle is simple: crosslinks are generated in a small space (intra-group), but are searched in a large space (inter-group); therefore, any identified peptide pair crosslink is considered plausible if they are in the same peptide group, and considered non-crosslinkable otherwise. This provides an FDR estimation independent of TDA, and hence can serve as a validation of the latter. Beveridge-NC-2020's method is limited

by its higher cost of synthetic peptide crosslinking and poor scalability, and hence cannot be used to mimic complex samples.

This manuscript proposes an experimental method that can be considered as a natural and even perfect extension of Beveridge-NC-2020 to complex samples. The design procedure is quite similar: Divide the *E. coli* sample into 44 SEC fractions, crosslink among proteins in each fraction, then pool the fractions together for MS sampling, identify crosslinks by searching the whole database of *E. coli*. Again, crosslinks are generated in a small space (intra-fraction), but are searched in a large space (inter-fraction). Therefore, any identified peptide pair crosslink or the corresponding PPI is considered plausible if the crosslink related protein pair is in the same fraction, and considered non-crosslinkable otherwise. This provides an FDR estimation independent of TDA, and hence can serve as a validation of the latter. In particular, this method can be applied to complex samples.

The method is applied to validating the two propositions with xiSearch + xiFDR, with largely successful results:

For Proposition 1: If 5% TDA FDR is controlled at levels lower than PPI, either separating or combining self and heteromeric crosslinks, then the estimated PPI FDR by the new experimental method will be at least over 10% (heteromeric RP) and as large as over 35% (naïve CSM), significantly higher than the target 5% PPI FDR.

For Proposition 2: If 5% TDA FDR is controlled at the PPI level, then the estimated PPI FDR by the new experimental method will be 6.6% (BS3) or 4.9% (DSSO), only a slight deviation from the target 5% PPI FDR.

This manuscript also proposes a trap-database method, another validation method in the same spirit of Beveridge-NC-2020 and the experimental method of this paper: generate crosslinks in a small space (the original database) and search crosslinks in a large space (the original database plus a trap database); crosslinks within the small space are plausible, and those outside are non-crosslinkable. Therefore, this trap-database method is also independent of TDA and can be used to validate the TDA FDR, and the validation results are similar to those of the experimental method as listed above. In particular, the trap-database method is purely computational, and hence can be more widely used than the protein fractionation based experimental method, since the latter can only be used with its specific data set, while the former can be used with arbitrary data sets. However, the trap-database method can be considered as having only one fraction, and hence should be weaker than the multi-fraction based experimental method in terms of discriminating power, which is a conjecture to be proved.

In short, this manuscript contributes two TDA-independent validation methods, one is experimental, and the other is computational, both of which can be used to estimate crosslink FDR at all levels and for complex samples. The two methods share the same design principle, i.e. to generate in a small space and to validate in a large space. There are other validation methods for complex samples, e.g. the 15N-labeling method in Chen-NC-2019, the multi-fragmentation method in Zhao-JPR-2020, which share a different principle: to provide more positive control information for the target, rather than to provide more negative control information as the two methods proposed in this paper. All these methods are shaping the crosslinking MS towards the right direction.

Major Concerns

There are five occurrences of “reliable” or “reliably” in this manuscript:

- Title: “Reliable identification of protein-protein interactions by crosslinking mass spectrometry.”
- Abstract: “Using a carefully controlled large-scale analysis of *Escherichia coli* cell lysate, we demonstrate that false-discovery rates (FDR) for PPIs identified by crosslinking mass spectrometry can be estimated reliably.”

➤ *Results: “In contrast, first merging CSMs for each PPI and then assessing the FDR gave more reliable results: 6.6% and 4.9% false PPIs when applying 5% PPI-FDR (Fig. 2e) for BS3 and DSSO, respectively.”*

➤ *Discussion: “In this work, we experimentally demonstrated that Crosslinking MS can reliably identify PPIs.”*

➤ *Discussion: “Crosslinking MS for mapping PPIs now has a reliable FDR estimation procedure.*

Question: How **reliable** is the TDA FDR control of xiFDR?

*Does it mean that the PPI FDR estimated by xiFDR will deviate from the true value by no more than 2% absolutely? Does it mean that the PPI FDR estimated by the protein fractionation based experimental method is **reliable**?*

*To trace back, how **reliable** are TDA FDR controls in routine protein identification?*

I do not think that TDA FDR controls in routine protein identification are reliable:

(1) The TDA FDR model can only model random matching errors, but cannot model homologous matching errors. That is, the model contains a systematic error.

(2) The TDA model has a so-called 1:1 hypothesis, i.e. a random matching will match targets and decoys with the same probability. However, there are several ways to construct decoys, e.g., reversing, shuffling, or reversing with local swapping (“the enzyme specific amino acids were swapped with their preceding amino acid” as used in MaxQuant and this manuscript). Nobody knows whether the 1:1 hypothesis will be satisfied in practice, given a target sequence database coupled with a decoy construction method, given a mass spectrometer and a search engine.

(3) Even if all spectra are matched to the correct peptides, further mapping peptides to proteins is not simple. Shared peptides will result in different ways of protein grouping. Very very large data sets will produce an anomaly under the classic TDA FDR model: more data result in less results.

Although I do not trust TDA FDR methods, I support their use since they are simple and universal, and in particular, no other methods are more reliable and meantime as simple and universal. However, currently computational proteomics bases its precision solely on the TDA-FDR model, which is quite dangerous to me.

We understand these relevant concerns of the reviewer. We sought to address this and later comments by generating more negative controls that test how accurate the decoy-based FDR is. As you can see from Figure S2 using four control types, “non-crosslinkable”, (multiple) entrapment, wrong precursor and wrong crosslinker mass, the estimation of FDR by the target-decoy approach – when done as we suggest in our work – is surprisingly accurate given the limitations listed above. This was a relief to us and, we assume, for the field.

Crosslinking MS is more complex than routine protein identification, and hence its TDA-FDR control cannot be simpler, and should be used with greater caution. While I fully support Proposition 1, since it has strong evidence both inside and outside this manuscript, I am deeply concerned that both the authors and the readers may take a simplistic view of Proposition 2, for which the evidence is not strong enough and may never be. I suggest the authors gather new evidence to achieve a comprehensive understanding of Proposition 2, and discuss limitations of the two methods. Even if the new evidence does not support Proposition 2 as the available evidence in this manuscript did, I still support the publication of this manuscript and the adoption of xiFDR.

Suggestion 1. *Use at least another crosslink search engine to test the two propositions and the two validation methods proposed in this manuscript.*

The two validation methods are universal, similar to the method in Beveridge-NC-2020. However, Beveridge-NC-2020 tested its method with five search engines including xiSearch, Stavrox/Merox, XlinkX, pLink, Kojak and two variants for Kojak and for Merox. In comparison, this manuscript only tested one search engine xiSearch. This manuscript claims that xiFDR v2.0 is crosslink search software independent, which needs direct evidence.

The xiFDR tool is independent of xiSEARCH. xiFDR uses the field standard mzIdentML format as input. As long as the search output of a given search engine complies with that standard data format, xiFDR will work with it. Note that xiFDR must be supplied with decoy matches from the search tool. Importantly, these decoys must model the false targets.

Following the suggestion of the reviewer we searched our dataset additionally with pLink2 and Merox (only DSSO data for Merox, as Merox supports large databases only for cleavable crosslinkers). Since neither Merox nor pLink2 currently export unfiltered lists of targets and decoys as mzIdentML it is currently not possible to input them to xiFDR directly.

We therefore wrote customized python scripts to convert the tables from these software to a xiFDR compliant format. We extracted the unfiltered output (targets and decoys) from these search engines and, using xiFDR, calculated the decoy-based FDR and decoy-independent error on CSM- and PPI-level. To our surprise, the xiFDR calculated decoy-based FDR did not match the decoy-independently determined errors at CSM level. This caused us concern so we decided to check if the search softwares pLink2 and Merox were producing decoys that truly represented random search space.

To test this we used the inbuilt FDR functions of both Merox and pLink2 to generate their standard output and then determined the observed error based on our decoy-independent controls. Applying pLink's 5% heteromeric decoy-based CSM-level FDR, the observed error of heteromeric CSMs was found to be 17.0% (BS3) and 30.1% (DSSO) for the non-crosslinkable control and 17.8% (BS3) and 18.1% (DSSO) for the human entrapment control (Reviewer response figure 1a). In Merox, these errors were 15.5% for the non-crosslinkable control and 42.0% for the human entrapment control.

An entrapment database is a known wrong search space, like a standard decoy database, that is unknown to the search engine. Therefore, matches to the human entrapment database and matches to the decoy database should have very similar score distributions. For xiSEARCH, these distributions are comparable (Reviewer response figure 1b). With the assumption that matches to the entrapment database represent random matches, it is obvious that neither pLink2 nor Merox output decoys that model the random search space / false positives (Reviewer response figure 1c,d).

We hypothesise that these discrepancies may come from improper decoy generation, internal rescoring of target and decoy matches by machine learning, or sub-score thresholding, but we cannot say for sure without access to the code of pLink2 and Merox. This is further supported by the fact that mixed *E. coli* / human TD matches behave like *E. coli* TD matches for all three search engines, i.e. Merox and pLink2 but not xiSEARCH treat decoy matches differently from target matches (Reviewer response figure 1b,c,d, right). Since xiFDR relies on the decoys returned by the search engine, it was therefore not possible to calculate a reliable PPI FDR for these search engines.

Reviewer response figure 1: Comparison of decoy-based FDR with observed error for three search engines, xiSEARCH, pLink2, and Mercox. a) Observed errors of heteromeric CSMs at 5% decoy-based heteromeric CSM-level FDR in pLink2, Mercox and xiSEARCH when using the non-crosslinkable and human entrapment controls. b, c, d - left) Score distributions of target-target matches to *E. coli* peptides (plausible matches), mixed *E. coli* - human target-target matches (false matches) and corresponding target-decoy matches (*E. coli* & *E. coli* - human). b, c, d - right) Close-up. The score distribution of three groups of known false matches,

namely mixed *E. coli* - human target-target matches, mixed *E. coli* - human target-decoy matches and *E. coli* target-decoy matches, look very similar for xiSEARCH. For pLink2 and Merox only the two target-decoy match distributions look similar, while the mixed *E. coli* - human target-target distribution deviates from the other two distributions of false matches. As the search engines are aware of the decoys but not of the false targets from the entrapment database, this indicates mistakes at the step of decoy generation or decoy handling in pLink2 and Merox. One consequence is that the FDRs reported by pLink2 and Merox, respectively, are underestimating the actual errors. This leads to inflated search results at the reported, but wrong, FDR and thus to less reliable findings.

Suggestion 2. Besides PPI FDR, validate FDRs at lower levels including CSM, PP and RP, and besides FDR for heteromeric crosslinks, validate FDR for self crosslinks.

FDRs at any information level (CSM, PP, RP and PPI) or type (self, heteromeric) may be useful and necessary. However, in this manuscript, only PPI FDR of xiFDR is validated by the two validation methods, although it seems that the same methods can be used to validate TDA FDRs at all other levels. Now that this manuscript claims that the PPI FDR can be estimated reliably, I think most readers may take it for granted that FDRs at all other levels can be estimated reliably, too. Please prove or disprove this impression.

We included an estimation of error for the lower heteromeric-FDR result levels (see Fig. S3) and added a comment to the main text: *Note that this also holds true for other reporting levels (i.e. CSMs, peptide pairs and residue pairs) (Fig. S3).*

Regarding performing this analysis focused on self-links, the larger the database becomes, the less likely randomly matching self-links become (as random matches are much more likely to fall into the larger search space of heteromeric links). At the large database sizes used here, random self-links are very rare and we do not reach the desired FDR. For example, for 5% heteromeric CSM-FDR, we only reach 0.2% and 0.4% for BS3 and DSSO self-links, respectively.

Suggestion 3. Try another way to estimate the FDRs by the two validation methods with the *E. coli* data set.

Let's discuss a general framework for error estimation by a validation method. Please refer to (1) Chen-NC-2019 (<https://doi.org/10.1038/s41467-019-11337-z>), Supplementary Note 3, Deducing an NaN-FDR for a search engine independent of the TDA-FDR.

(2) Zhou-JPR-2019 (<https://doi.org/10.1021/acs.jproteome.8b00993>).

Given a set of N identifications of any kind, with T correct and F incorrect. Hence $N = T + F$, and the error rate is F/N .

*Given a validation method with a false positive rate e_1 and a false negative rate e_2 for the data set. The validation method detects S suspects from the N identifications; specifically, it detects $T * e_1$ false alarms and $F * (1 - e_2)$ true alarms.*

*Hence $S = T * e_1 + F * (1 - e_2)$.*

By simple deduction, $F/N = (S/N - e_1) / (1 - e_1 - e_2)$.

The ratio F/N is the error rate to be estimated, denoted by $Error_{estimated}$. And the ratio S/N is the detected error rate, denoted by $Error_{detected}$. Then $Error_{estimated} = (Error_{detected} - e_1) / (1 - e_1 - e_2)$.

For the validation method in Beveridge-NC-2020 and the two validation methods in this manuscript, they are all based on the entrapment validation approach, and it can be reasonably assumed that $e_1 \approx 0$, while e_2 may be quite different for each method. Then $Error_{estimated} = Error_{detected} / (1 - e_2)$.

Compare the above general formula with the specific formula of the experimental validation method:

$$Error_{PPI, non-crosslinkable} = Error_{estimated},$$

$$\frac{n_{false}}{n_{plausible} + n_{false}} = Error_{detected},$$

$$\frac{KR_{plausible} + KR_{false}}{KR_{false}} = 1.09 = (1 - e_2)^{-1}.$$

The formula of the trap-database method can be interpreted similarly.

Relatively speaking, $Error_{detected}$ can be calculated objectively, and whether $Error_{estimated}$ is reliable depends on whether e_2 is estimated reliably. This is why I trust Proposition 1 more than Proposition 2, since Proposition 1 can still be proved with no need of estimating e_2 . Specifically, naïve or heteromeric 5% FDR at lower levels will still result in $Error_{detected} \gg 5\%$ at the PPI level. No matter how e_2 is estimated, $e_2 \geq 0$, and hence $Error_{estimated} \geq Error_{detected}$. Therefore, $Error_{estimated} \geq Error_{detected} \gg 5\%$.

In contrast, Proposition 2 may no longer be true if the estimation of e_2 (the equivalent of 1.09 or 1.998) is not reliable. As stated in the manuscript, 11% of the proteins found in 590 PPIs at 1% PPI FDR had no self-links, which is a sign to me that the actual FDR may be higher than the expected 1%. Although 1.09 and 1.998 are reasonable and clever estimations, I think it might be more convincing if another independent estimation of e_2 can be tried and hence another independent estimation of FDR can be obtained.

The *pValid* paper Zhou-JPR-2020 might be a reference. Shift the precursor masses by some Daltons, then try the same search process as before. All search results shall be incorrect, among which some are plausible, and the other are non-crosslinkable. You can estimate e_2 at every level from CSM to PPI, and e_2 may have a different value at different level, rather than the rigid 1.09 or 1.998.

We have added two additional controls employing a similar approach as proposed by the reviewer. For one, we have searched the data with a wrong-mass crosslinker (see methods). Secondly, we have used a similar approach as suggested by the reviewer. Spectra passing a 1% heteromeric CSM FDR were searched as well as the same spectra with two different wrong precursor masses (see methods). This control does not need to be normalized as there are negligible amounts of unknown false positive matches (only 1%, given the previous FDR). These two controls agreed with our previous controls (see new Figure S2), showing that the normalization we employ is correct.

For the trap-database method, actually you can use arbitrary databases as entrapment, which is no longer required to be similar to the original database, as long as you can estimate the corresponding e_2 .

We have added two extra entrapment databases to our control (Figure S2). The full proteome of *S. cerevisiae*, leading to a factor of 1.19, as well as the *S. cerevisiae* proteome combined

with our previously used human proteins, with a normalization factor of 1.16. All three entrapment databases lead to a similar error. This supports our normalisation approach. This does trigger a note of caution. If one uses entrapment databases without normalisation one will underestimate the actual error (which is done by Götze et al. <https://doi.org/10.1021/acs.analchem.9b02372>). The smaller the entrapment database is, the larger is the underestimation.

There is no free lunch. For Beveridge-NC-2020, it seems that $e_2 = 0$, and the error rate can be estimated more objectively and straightforwardly ($r_{roestimated} = Error_{detected}$), but the method cannot be extended to large scale or used for complex samples. For the two methods proposed in this manuscript, they are applicable to complex samples, but they have to estimate e_2 , a key parameter of the validation method.

Anyway, large-scale benchmark data sets or validation methods are difficult to design, yet they are indispensable to precision proteomics, including precision crosslinking proteomics. This manuscript made a solid step forward in the right direction. While the TDA FDR model will accompany our journey, it is only a convenient walking stick, and cannot be solely relied upon.

Minor Concerns

➤ *General:*

(1)

The original version of this manuscript, posted in bioRxiv, was apparently prepared as Brief Communications, and hence the statements and the references were concise. But now, the manuscript was prepared as Article, the statements in the text and figure legends and the references should be enriched accordingly.

We have added references now and amended text and figure legends (see responses to specific comments below).

(2)

Please add numbers for all bars in all bar chart displays, including Fig. 1b, Fig. 2, Supp Figs. 1b and 1c (the two dotted lines), Supp Figs. 2, 3, 4. Additional tables may be used when necessary.

We have added numbers to Figures 1b, Figure S1b,c. Source data are now included according to the Nature Communications standard.

➤ *“This target-decoy approach has been adapted for Crosslinking MS14–17.” Yang-NM-2012 (<https://www.nature.com/articles/nmeth.2099>) should be cited. See its Abstract (“pLink reliably estimates false discovery rate in cross-link identification...”), its text (“The FDR calculation is based on in silico cross-linking of forward (F) and reversed (R) peptide sequences, computed as the number of identified F-R and R-F cross-links subtracted by the number of identified R-R cross-links, then divided by the number of identified F-F cross-links (details in Supplementary Note).”), and its Supplementary Information (Supplementary Figure 15 “FDR estimation” and Supplementary Note “Estimation and control of FDR” at page 43).*

We have added this citation to the introductory sentence, “This target-decoy approach has been adapted for Crosslinking MS”.

➤ *“The first, whether to consider crosslinks between peptides within one protein sequence (self-links, including homomeric crosslinks) separately from crosslinks between protein sequences (heteromeric crosslink)15.”*

Beside your reference 15, your reference 4 or Chen-NC-2019 should also be considered as reference, which conducted both theoretical and experimental studies concerning separate control, as shown explicitly in its text and supplementary information: "Several studies have shown that the use of separate control of the FDR for inter-protein and intraprotein identifications, rather than global control, is an effective means of improving credibility of inter-protein results^{11,12}. Our data analysis confirmed this and found that intra-protein PSMs and inter-protein PSMs show different changes when switching from global FDR control to separate FDR control (Fig. 3d, e). For intra-protein PSMs, more results were reported under separate FDR control and its percentage of NaN ratios was only slightly higher than that under global FDR control (Fig. 3d). For interprotein PSMs, many fewer results were reported under separate FDR control and its percentage of NaN ratios decreased notably (Fig. 3e). This phenomenon also existed in the results of Protein Prospector (Supplementary Fig. 8), which recommended separate FDR control in its study¹². Furthermore, our theoretical analysis about the relationship between the global FDR and subgroup FDRs of intra-protein and inter-protein identifications was accordant with the above experimental phenomenon; please see Supplementary Note 2 for details." The reference 11 therein is your reference 15, and reference 12 therein or Trnka-MCP-2014 should also be considered as your reference.

We have included this reference to the introductory sentence "The first, whether to consider crosslinks between peptides within one protein sequence (self-links, including homomeric crosslinks) separately from crosslinks between protein sequences (heteromeric crosslink)".

➤ "In our data, the chance of matching a decoy crosslink (random) within the heteromeric crosslinks is 10.6 times higher than within the self-links (Fig. 1b). Controlling FDR in the total set of CSMs, and then selecting only heteromeric matches thus enriches for false positives. This leads to a large underestimation of the error within heteromeric CSMs, which describe PPIs (Fig. S1). Consequently, heteromeric crosslinks must be considered separately from self-links during FDR estimation.

First, under what condition was 10.6 obtained? There was no explanation, either in the text or in the figure legend.

We have now added the information of the database size to the sentence and corrected "PPIs" to "CSMs" in the figure legend.

Second, do you mean that the statement of the first sentence is a sufficient condition for the statement of the second sentence, and further for the third sentence? Suppose that there were ten times fewer self-links in the sample, did your statements and deductions still hold true? Please refer to "Supplementary Note 2. The relationship between the global FDR and the subgroup FDRs of intra-protein and inter-protein cross-linked identifications" of Chen-NC-2019 (your reference 4).

This experiment is not about absolute numbers in the database but chance of random matching within each group. To be independent of total group size, we separately sampled the same number of CSMs for self and heteromeric links.

There might be construed hypothetical cases where not separating self-links and heteromeric matches for FDR leads to the same result as separating them. However, our theoretical considerations mandate the separation if one aims for a robust procedure.

➤ Of note, in our experimental control 87% of false PPIs involved proteins that were seen only with heteromeric crosslinks, i.e. that lacked self-links (Fig. 2c). In the entrapment control this number increased to 100% (Fig. 2c).

How were the false PPIs obtained? Were they obtained by first filtering CSMs at a naïve 5% CSM-FDR, then merging the heteromeric CSMs into PPIs, and finally classifying the PPIs into false (non-crosslinkable) and true (plausible)?

Yes, this is correct. We clarified it in the text.

"However, our experimental control revealed the problem of a 'naïve' CSM-level FDR for reporting PPIs. The heteromeric CSMs of 'naïve' 5% CSM-FDR were merged into PPIs. This led to 36% of PPIs being false in the DSSO dataset (Fig. 2a). In the BS3 dataset the results were very similar with 'naïve' 5% CSM-FDR leading to 35% false PPIs (Fig. 2b)."

➤ *If observed at all, 'heteromeric only' proteins had a lower median abundance than all proteins in the sample suggesting that they are enriched in random matches (Fig. 2c). Error: Fig. 2c should be Fig. 2d.*

The figure calling has been corrected.

Again, how did you obtain the " 'heteromeric only' proteins" and "all proteins"? At naïve CSM-FDR or at PPI-FDR? 1% or 5%? With the E. coli proteome database of 4350 or 1926 proteins? With additional pruning of non-crosslinkable PPIs? In particular, how did you obtain the self-only proteins? These proteins should only come from self crosslink CSMs, but you did not clearly define how to estimate and control the FDRs for self cross links at various levels, from the CSM level to the protein levels.

We have clarified the database now in the beginning of the section: *"We first searched against a database comprising all (4350) E. coli proteins, including those not detected in our sample."*

We have also clarified the FDR approach and merging to PPIs (see comment above).

We have not pruned the PPIs.

Self-only proteins are proteins that are not part of any (heteromeric) PPIs. xiFDR considers these separately when computing PPI FDR as "self-PPI" group.

➤ *As predicted, CSMs rarely corroborated each other in false PPIs while plausible PPIs were supported by multiple CSMs (average 1.2 versus 4.6 CSMs), irrespective of the crosslinker (Fig. 2a,b).*

For an Article in Nature Comm, there is enough space for complete description. Hence I suggest that besides describing 1.2 and 4.6 for BS3, 1.3 and 5.2 for DSSO should also be added. Furthermore, the explanations in the text should be added to the legends to Fig. 2a and 2b.

We now give the exact numbers in the text and we have added the sentence to the figure legend.

➤ *In contrast, first merging CSMs for each PPI and then assessing the FDR gave more reliable results: 6.6% and 4.9% false PPIs when applying 5% PPI-FDR (Fig. 2e) for BS3 and DSSO, respectively. This also applies to other FDR thresholds and the entrapment control which indicated an error of 1.4% for BS3 and 3.6% for DSSO when applying 5% PPI-FDR (Fig. S2).*

If no iBAQ control is applied for non-crosslinkable PPI judgement, what are the numbers in Fig. 2e?

We attached the errors for the control without an iBAQ cutoff as a table below. However, we think the iBAQ cutoff is essential, as a protein with very low abundance is easily detected in regular proteomics but very rarely seen in crosslinking (see Figure S1b). Anyway, the requested numbers are:

level FDR	grouping	error BS3	error DSSO
CSM	no	28.2%	29.7%

Pep. Pair	no	23.9%	23.2%
Res. Pair	no	20.4%	20.7%
PPI	no	6.1%	4.8%
CSM	yes	11.6%	11.3%
Pep. Pair	yes	10.0%	9.5%
Res. Pair	yes	9.3%	8.2%
PPI	yes	3.2%	2.1%

There is no such error information as 1.4% or 3.6% in Fig. S2.

Figure S2 is now simplified. The exact errors are now included in the source material, according to the Nature Communications standard.

➤ *In previous studies, the quality of identified crosslinks was assessed by measuring interresidue distances in known protein structures. However, for proteome-wide crosslinking studies, this approach is inherently biased towards true interactions as 'real' crosslinked matches are likely to be enriched in these known complexes. The majority of random PPIs are neglected by this FDR evaluation method, making this approach completely inadequate for reliable PPI error estimation.*

The sentence in the middle is hard to understand.

We changed the sentence and bolstered it with a citation. The new sentence is:

“However, for proteome-wide crosslinking studies, this approach is inherently biased towards true interactions as they are likely to be enriched in known complexes.”

➤ *Figure 1: Considerations for crosslinked PPI-FDR and experimental workflow. b) Fraction of decoys in 10 random picks of 100 self and 100 heteromeric PPIs from the search output before FDR filtering. Please complement the bars with numbers, and describe more exactly how to obtain the results, e.g. before FDR filtering at the PPI level yet after FDR filtering at the RP level 5%? Or before no FDR filtering at any level and after merging all top-1 CSMs into PPIs?*

We corrected “PPIs” to “CSMs” in the figure legend and added “any FDR filtering”. Numbers were added to the bars.

➤ *Figure 2: Comparative analysis of different methods of FDR estimation in Crosslinking MS.*

Please complement all bars with the corresponding numbers. Please explain all the numbers in the figure, such as 5.2 and 1.3 in a), 4.6 and 1.2 in b), 339 and the other 5 numbers in d).

Tables of the source data have been provided according to Nature Communications requirements for Figure 2a,b,d,e,f. We added “n = “ to Figure 2d.

Please explain the inconsistency between the numbers in Fig. 2d and Supp Fig. S4C.

Fig. S4 relates to the final PPI network, using heteromeric PPI-FDR at 1%. In contrast, Fig. 2d employs naive CSM-FDR at 5%. Therefore we expect many wrong PPIs in the latter data, while the 1% heteromeric PPI-FDR data should be much more reliable. Note that the number of “heteromeric only” proteins decreases dramatically.

“Average of BS3 and DSSO data (separated in Fig. S4c): “Fig. S4c” should be “Fig. S3, a and b.”

We changed the figure legend accordingly.

- *LS-MS protein ID: ACG target set to 1*105.*
- *LC-MS for crosslink ID: ACG of 5*104 ACG should be AGC.*

We corrected the spelling accordingly.

- *Crosslink database search*

“Matches were not filtered for having DSSO-specific signature peaks.”

What does this mean? Where are the differences between the algorithms of xiSearch-DSSO and xiSearch-BS3?

The only difference between xiSearch-DSSO and xiSearch-BS3 is the definition of the crosslinker cleavage and therefore additional fragments. We have added the sentence “XiSearch algorithms are identical for both crosslinkers (BS3 and DSSO).” to the methods.

After the search, results might be further filtered to spectra containing the respective peptide signature peaks / doublets, which was not done here to keep DSSO and BS3 as comparable as possible.

“Crosslink sites for both reagents were allowed for side chains of Lys, Tyr, Ser, Thr and the protein N-terminus.”
What is the proportion of Lys-Lys crosslinks in self crosslinks and in heteromeric crosslinks at each level of separate FDR control at 5%?

Please see the table below.

Result level	FDR level	% K-K self (BS3 DSSO)	% K-K heteromeric (BS3 DSSO)
CSMs	CSM	79.8 86.2	81.2 91.1
Peptide pairs	Peptide pair	77.5 83.8	80.4 90.9
Residue pairs	Residue pair	75.3 81.6	80.2 91.3

How many MS/MS spectra are sampled for crosslink identification?

The 7,287,062 acquired MS spectra were all searched.

What is the MS/MS identification rate with xiSearch+xiFDR on your data set, at 5% CSM-FDR separate for self links and heteromeric links?

The identification rate is 18% for DSSO and 19% for BS3.

Are the contamination proteins taken into database search?

No.

“The same database as for the non-crosslinked samples was used. For the entrapment database control the database was extended by the same number (4,353) of human proteins.”

The number 4353 was inconsistent with the number 4350 used before in “Quantitative proteomics database search”.

We thank the reviewer for noticing this. It is in fact only a typo and we corrected the text accordingly.

“Decoys were generated for all searches, including the entrapment database. For this, protein sequences were reversed and for each decoy protein the enzyme specific amino acids were swapped with their preceding amino acid.²⁵”

Does this simple swapping operation result in notable effect on the identification?

We have not done a detailed study of the effect but previous anecdotal evidence shows that this can prevent matching reversed peptides of exact same composition with almost the same score as the presumed correct target peptide. One could argue that this introduces an artificial difference between false target and decoys - but our data here show (as in they make sense) that if such a difference exists it is below the level of precision that we can reach with a decoy-based FDR.

➤ *FDR calculation*

“Results were filtered prior to FDR to crosslinked peptide matches having a minimum of three matched fragments per peptide, a delta score of 15% of the match score and a peptide length of at least six amino acids.”

In your own paper or your reference 6, your criteria are: “Cross-links between two different proteins were analyzed with the following parameters: prefilter cross-links only, Δ score 0.5, minimum number of fragments per peptide five, with eight amino acids as minimum peptide length.”

Why do you select different criteria for two papers, according to what principles?

The optimal delta score and number of fragments per peptide is different depending on the search and sample. Factors influencing it are for example database size, sample and search complexity, quality of the spectra or crosslinker. Newer versions of xiFDR include an optional optimization on both variables. However, in this manuscript we did not use this optimization as to keep the different FDR calculations comparable.

“Additionally, identifications ambiguously matching to two proteins or more were removed.”

Among the peptides pairs at 5% peptide pair FDR, what is the proportion of the ambiguous?

The proportion is 1.8% for DSSO and 1.9% for BS3.

“FDR was calculated based on decoy matches by xiFDR (version 2.0dev) using the formula¹⁶: $FDR = (TD - DD) / TT$. Depending on the experiment, FDR was employed on different result levels (CSM, peptide pair, residue pair or protein pair) with defined thresholds.”

In routine protein identification, the TDA model has the 1:1 hypothesis for random matches in T and D; based on that, in crosslink identification, the TDA model has two hypotheses for random matches, the first is the 1:2:1

hypothesis for the random-random crosslinks in TT, TD and DD, and the second is the 1:1 hypothesis for the random-correct crosslinks in TT and TD; based on the two hypotheses, the crosslink TDA formula $FDR = (TD - DD) / TT$ can be deduced.

There was one validation experiment for the 1:2:1 hypothesis at CSM level in Yang-NM-2012, Supp Fig 15B. Could you provide a similar validation experiment for the 1:2:1 hypothesis at other levels higher than CSM, especially at the PPI level? For example, delete the 1926 proteins from the 4350 proteins, and use the remaining proteins to search the E. coli data set, merge all top-1 crosslink CSMs to higher levels. Your human entrapment database can also be used directly for such a validation.

We took “human only” identifications at 100% FDR to evaluate how the number of TT, TD and DD matches relate to each other. In fact, at PPI level, 12902 human TT PPIs, 25357 human TD PPIs and 12068 human DD PPIs are identified for DSSO (3247, 6220, 2979 for BS3), leading to a ratio of 1:1.97:0.94 for DSSO and 1:1.92:0.92 for BS3.

“Depending on the experiment, FDR was employed on different result levels (CSM, peptide pair, residue pair or protein pair) with defined thresholds.”

For the CSM level, are the CSMs allowed to be redundant or required to be unique? There are two cases of “unique CSMs” occurred in this manuscript, and hence maybe you allow both types. For some search engines, “CSMs” are redundant by default, unless “unique CSMs” are used. There is also one case of “unique residue pairs” in this manuscript.

In this manuscript, only unique CSMs were used (which is also the default in xiFDR), since redundant CSMs can heavily skew the FDR calculation and should not be used. We have now added redundant CSMs as an extra level of FDR in Figure S2 and added a clarification to the text: “Note that CSMs in this manuscript refer to unique CSMs, using redundant CSMs will produce spurious FDR estimations (Fischer and Rappsilber 2017 PMID: 28267312) (Fig. S2).”

“For the final PPI network, BS3 and DSSO PPIs were separately filtered to 1% heteromeric PPI-FDR. As the scores cutoff differed between the two datasets, the score was normalised to range between 0 and 1 for subsequent joining of the two lists. The combined list was then filtered again to 1% FDR.”

Please describe how to do normalization, joining and further FDR control.

We have changed it to: “For the final PPI network, BS3 and DSSO PPIs were separately filtered to 1% heteromeric PPI-FDR. As the score cutoffs differed between the two datasets, the scores from each dataset were first normalized (i.e. the local FDR was used as a normalized score) to range between 0 and 1. Subsequently, the two tables were concatenated and the FDR calculated again as described above and filtered to 1% FDR.”

➤ *Non-crosslinkable control: The question now is what abundance is sufficient. As a heuristic, we removed the lower 5% iBAQ values (iBAQ of $4.3E6$). This removed very few (169, 7%) of our identified protein pairs (n.b. at a very loose FDR threshold), i.e. did not change much the outcome of our identification data by generating false negatives. // For any identified protein pair to be considered possible, both proteins had to be found in at least one SEC fraction together with individual iBAQ values above our iBAQ threshold (iBAQ of $4.3E6$). Else, they were defined as non-crosslinkable. 544,274 (6% of all theoretically possible PPIs in the E. coli proteome) PPIs are defined as plausible (see Table S4), while 8,914,801 (94%) PPIs are non-crosslinkable.*

First, contrast this with Supplementary Fig. 1: “Non-crosslinkable control definition. b) Abundance distribution of proteins identified with heteromeric crosslinks at a generous 10% heteromeric peptide pair FDR (best lower

iBAQ, see Methods). Protein pairs are accepted as plausible if both proteins reach the 5-percentile in the same fraction, otherwise the pair is defined as 'non-crosslinkable'.

Is the iBAQ threshold fixed or dynamic? The threshold is fixed at 4.3E6 across all SEC fractions, or dynamic at 5% in each SEC fraction?

The threshold is fixed across all fractions.

Second, if no such iBAQ control is applied, what will Fig. 2e will be?

See above.

Third, I suggest that you use "in the E. coli proteome of 4350 proteins" to provide more information for readers to verify your calculated ratios 6% and 94%.

We changed the text accordingly.

➤ *Supplementary Figure 1: Non-crosslinkable control definition. a) Illustration of the distribution of decoy matches for self and heteromeric crosslinks. If considered together for FDR calculation, heteromeric decoy matches will be matched more frequently and make up most of the summed decoy matches. If subsequently heteromeric matches are evaluated separately (e.g. for reporting PPIs), their error will be larger than the previously calculated FDR (which would only be correct for the data as a whole). b) Abundance distribution of proteins identified with heteromeric crosslinks at a generous 10% heteromeric peptide pair FDR (best lower iBAQ, see Methods). Protein pairs are accepted as plausible if both proteins reach the 5-percentile in the same fraction, otherwise the pair is defined as 'non-crosslinkable'. 544,274 (6% of all possible PPIs in the E. coli proteome) PPIs are defined as plausible (see Table S4), while 8,914,801 (94%) PPIs are non-crosslinkable. First, in a), in the FDR filtered results, the numbers of self targets and heteromeric targets seem to be equal, but in practice the number of self targets should be significantly larger than that of heteromeric targets. Why not use numbers of a real instance in your paper for such a demonstration?*

The figure is meant to be illustrative. Real instances can vary largely depending on database and sample.

Second, do you mean that "If considered together for FDR calculation, heteromeric decoy matches will be matched more frequently and make up most of the summed decoy matches." will be sufficient to guarantee "If subsequently heteromeric matches are evaluated separately (e.g. for reporting PPIs), their error will be larger than the previously calculated FDR (which would only be correct for the data as a whole)." ?

Yes.

Third, I suggest that you use "in the E. coli proteome of 4350 proteins" to provide more information for readers to verify your calculated ratios 6% and 94%.

We changed the text accordingly.

➤ *Supplementary Figures 2 and 3*

Please provide additional tables showing the actual numbers of all data points. You have pointed out that "at low FDR thresholds the calculated error is less reliable due to small numbers of false matches," and hence besides showing the FDR value, also indicate the number of decoys and/or the number of targets in the calculation of FDR. What is the minimal number of PPIs for a reasonable estimation of PPI-FDR?

Tables have been provided according to Nature Communications requirements. Fischer and Rappsilber 2017 (PMID: 28267312) proposed reporting the confidence interval of the FDR estimation. This might provide some guidance. However, “reasonable” is always in the eye of the beholder. Basically, it is similar to the question what FDR is acceptable. The more target matches one has the less impact does the stochastic nature of decoys have on the outcome. One possible approach is to take the apparent resolution - as in what would be the FDR if I had seen one decoy more or less - as a reference. Then the person doing the experiment has to decide if they can live with that accuracy or not. The important point here is to be aware that precision and in result the accuracy of a decoy-based FDR strongly depends on the number of matches making the cut-off.

➤ *Supplementary Figure 4: Properties of final crosslink PPI network.*

Please complement the bar chart display with numbers, for readers' convenience and for easy correspondence between text and figures. Fig. S4c seems to be inconsistent with Fig. 2d: (1) In Fig. S4c, self-only crosslinked proteins had the lowest median abundance, while in Fig. 2d, heteromeric-only crosslinked proteins had the lowest median abundance ; (2) The numbers at the top of Fig. S4c, i.e. 644, 274, 34, do not match those of Fig. 2d.

See above.

REVIEWERS' COMMENTS

Reviewer #2 (Remarks to the Author):

Review of the revised manuscript Lenz et al., "Reliable identification of protein-protein interactions by crosslinking mass spectrometry"

I appreciate very much the authors' work on their revised version of this important manuscript. The authors have addressed satisfactorily the points that I made on their original manuscript. In particular, they show that the hitherto uncharacterized protein YacL interacts with RNAP and refined the location of the interaction by using SDA crosslinking.

I am happy to recommend that the revised version be published in Nature Communications.

Reviewer #3 (Remarks to the Author):

Swantje Lenz, Ludwig R. Sinn, Francis J. O'Reilly, Lutz Fischer, Fritz Wegner, Juri Rappsilber

Reliable identification of protein-protein interactions by crosslinking mass spectrometry

Submitted to Nature Communications, Revision

Review of Revision

I think the authors have done their best to address all of my concerns. I appreciate their efforts in addressing my three major concerns, and their gains are clear: they have gained increased confidence on their validation experiment design, on the validation software xiFDR, and on the search engine xiSearch. The manuscript can be published either as is or subject to some minor revisions as indicated below.

➤ Optional discussion of possible limitation/optimization of design/analysis of the fraction-based validation method

False discovery rate (FDR) control is a cornerstone of proteomics. While the target-decoy approach (TDA) becomes the method of choice for FDR control of mass spectrometry (MS) based protein and protein crosslinking identification, the rationality of TDA-FDR control is mainly based on vague assumptions, and its validation has always been a challenge due to lack of gold benchmark methods with scalability, credibility and easy annotation. For crosslinking MS, Mechtler's validation method (NatComm-2020), based on synthetic peptides crosslinked in a small space yet searched in a large space, is both credible and easy to annotate, but it can hardly scale up to complex samples and to the protein-protein interaction (PPI) level, which is why Rappsilber's validation method proposed in this manuscript stands out: for validation of crosslinking in a complex sample, at the PPI level, together with the other three levels below, intra/inter separated and multi-level TDA-FDR control is shown to be necessary, and meantime sufficient by xiFDR + xiSearch.

The authors' optimism is shown in words in the manuscript: "*In this work, we experimentally demonstrated that Crosslinking MS can reliably identify PPIs using the target-decoy approach as a quantitative error metric. This negates the need for any additional heuristics suggested by others*²⁷." And also in the rebuttal: "*As you can see from Figure S2 using four control types, 'non-crosslinkable', (multiple) entrapment, wrong precursor and wrong crosslinker mass, the estimation of FDR by the target-decoy approach – when done as we suggest in our work – is surprisingly accurate given the limitations listed above (proposed by the reviewer). This was a relief to us and, we assume, for the field.*" They are so optimistic that they actually do not discuss any limitation of their methods in Discussion.

Specifically, their optimism is based on the results shown in Fig. 2e, Supp Figs 2 & 3, which are assembled as follows:

Heteromeric FDR (%) BS3 DSSO (Fig 2e, Supp Figs 2&3)	Decoy-based xiFDR	Fraction-based Non-crosslinkable	Entrapment (E.coli + Human)	Entrapment (E.coli + Yeast)	Entrapment (E.coli +Human +Yeast)	Wrong precursor mass	Wrong crosslinker mass
PPI	5.0 5.0	6.6 4.9 *	1.4 3.6	6.1 4.2	5.5 4.2	6.2 7.0	7.4 5.3
Residue Pair	5.0 5.0	5.5 4.8					
Peptide Pair	5.0 5.0	5.3 4.4					
Crosslink-Spec Match	5.0 5.0	4.8 4.2					
* 3.2 2.1 if without iBAQ cutoff							

The data in the above table show really good consistency between the FDRs estimated by the decoy-based xiFDR and the FDRs estimated by several independent methods, especially Rappsilber's fraction-based validation method. However, this consistency is subject to several approximations.

To estimate the fraction-based FDR, the key is to define non-crosslinkable or false results. By experiment design, all identified inter-fraction crosslinks are false, and all identified intra-fraction crosslinks are true. But in the practice of the manuscript, identified intra-fraction crosslinks may still be considered false if the associated proteins did not pass a detection limit, i.e. an iBAQ cutoff. Heuristically, the iBAQ cutoff threshold is set at an abundance of 5% or 4.3E6, and this affects the results notably: with iBAQ cutoff, the estimated PPI-level FDR is 6.6% (BS3) or 4.9% (DSSO), close to 5% TDA-FDR; without iBAQ cutoff, the estimated FDR is 3.2% (BS3) or 2.1% (DSSO), away from 5% TDA-FDR. While the iBAQ cutoff is reasonable, the choice of specific threshold is arbitrary to a certain extent; furthermore, a non-crosslinkable result is defined (identified and quantified) by MaxQuant, and another linear peptide search engine might provide different results.

Similarly, for xiFDR, there is also a heuristic step of prefiltering crosslink-spectrum matches (CSMs): "*Results were filtered prior to FDR to crosslinked peptide matches having a minimum of three matched fragments per peptide, a delta score of 15% of the match score and a peptide length of at least six amino acids.*" The choice of parameter values differs in different papers, indicating something arbitrary that may also affect the final consistency.

In computational proteomics, all such approximations are understandable, and the overall consistency achieved is impressive. However, it should be confessed that in comparison, Mechtler's validation design seems simpler and more objective: all intra-group CSMs are correct, and all others are incorrect, with no additional thresholds or software involved. Therefore, at least when validating CSM- and peptide-pair-level FDRs, Mechtler's validation method seems more convincing than Rappsilber's validation method.

Even if there is perfect consistency between the FDR estimations of xiFDR-like TDA methods and other independent FDR estimation methods, is this consistent FDR equal to the true FDR? Probably not. We should be aware of the limitations of the TDA model, as I already pointed out clearly in the first-round review. Till now, routine proteome identification has no agreed TDA-FDR control at the protein level. Proteome crosslinking identification is more complex: even with state-of-the-art software xiSearch+xiFDR on the data set of the manuscript, the identification rate of tandem mass spectra is no more than 20%, as stated in the rebuttal. Therefore, we should be cautious when treating the PPI-level TDA-FDR for proteome-scale crosslinking MS. Rappsilber's validation method marks a significant step forward, but it will not negate the need for any new validation methods, including optimization of their own. xiFDR might be the best method at present to control FDR especially at the PPI level, but it will not negate the need for further validation and improvement.

➤ About the Reviewer Response Figure 1 in the rebuttal

The Reviewer Response Figure 1 in the rebuttal raises intriguing concerns. Since the data set of this manuscript has not been made available yet, and details about the data analysis procedure associated with the figure have not been adequately specified, further confirmation cannot be done until the manuscript is published.

One highly probable reason for the higher observed CSM errors of pLink2 and MeroX compared to xiSearch+xiFDR, as shown in the figure, is that *by default*, pLink2 and MeroX apply TDA-FDR control to *redundant* CSMs, while xiSearch and xiFDR apply the control to *unique* CSMs. After TDA-FDR control of pLink2, if those redundant CSMs are merged into higher levels with no further TDA-FDR control, a higher FDR will be induced, as already measured either by the ¹⁵N-labeling method of pLink2 (NatComm-2019) or by Mechtler's method (NatComm-2020).

We have tested pLink 2 with Mechtler's method, and found that (1) pLink2's TDA-FDR control at the redundant CSM level was effective at the same level; (2) after merging those redundant CSMs into unique CSMs, yet without further TDA-FDR control, then the calculated FDR at the unique CSM level was higher than the calculated FDR at the redundant CSM level; (3) conducting TDA-FDR control at the unique CSM level by a new version of pLink2 was effective at this new level. In particular, all these three facts are no surprise as expected. The new version of pLink2, to be made public soon, has been equipped with TDA-FDR control at higher levels (including the unique CSM level) and has passed validation of Mechtler's method. FDR control at the redundant CSM level will still be supported in pLink2, since it is still useful, e.g. when calculating the interpretation rate of tandem mass spectra.

Therefore, if the authors construed pLink2's output of redundant CSMs (after TDA-FDR control at 5%) as unique ones, calculated the FDR by their non-crosslinkable validation method (possibly with iBAQ cutoff and normalization), and drew the figure, then the higher FDRs shown in the figure would be no surprise. The authors can easily check this. To our knowledge, CSMs reported by MeroX were also redundant.

The authors indicated another hypothesis that the discrepancies between pLink2/MeroX and xiSearch shown in the figure may come from improper decoy generation, internal rescoring of target and decoy matches by machine learning, or sub-score thresholding. pLink2's TDA-FDR control at the redundant CSM level uses protein sequence reversal for decoy generation, Percolator-like re-ranking, and no sub-score thresholding (NatComm-2019). Percolator has been an established technique since its publication (NatMeth-2007), with little discussion seen yet about whether it misuses the target-decoy approach (I will contact the authors of Percolator later). In principle, Percolator will surely suppress decoy matches and meantime some (hopefully incorrect) target matches sharing similar features with the decoy ones, but some other target matches, incorrect yet with features distinct from decoy ones, may survive from the suppression. Anyway, decoy matches can only model certain kind of random target matches, but cannot model all incorrect target matches, especially homologous errors, which is why I always worry about the reliability of TDA-FDR control of any kind, including xiFDR and that of pLink2. For crosslinking TDA-FDR control, one more complex issue arises: besides random-random peptide-peptide matches, there are also correct-random peptide-peptide matches, and these two kinds of incorrect matches might have different score distributions. Anyway, necessary and sufficient conditions for a search engine to be eligible to work with xiFDR or any other TDA-FDR control are worth further investigation, which is often neglected by both regular proteomics and crosslinking proteomics.

➤ About iBAQ cutoff:

Please display several best-quality CSMs that are identified by xiSearch+xiFDR but considered non-crosslinkable due to iBAQ cutoff.

Please show the three percentages of unique CSMs at 5% TDA-FDR that are with iBAQ = 0, $0 < \text{iBAQ} \leq 5\%$ (4.3E6), and iBAQ > 5%, respectively.

- About normalization factor:
Please use more words to describe the principle and the method of calculating the normalization factor 1.09 for non-crosslinkable validation experiment at the PPI level. I am sorry I cannot understand yet.
- About lower-level validation:
“Supplementary Figure 3: Observed errors using the non-crosslinkable control on information levels lower than PPI-level.” There is no definition of “observed error” in the main text or supplementary information. There is only definition of *ERROR_{PPI,non-crosslinkable}* in Methods, but no similar definitions for any errors using the non-crosslinkable control on information levels lower than PPI-level. Are they entirely the same as *ERROR_{PPI,non-crosslinkable}*, including the same normalization factor 1.09?
- Type I or type II error:
“This approach assumes that the rate of matches to the decoy database is an estimator of false positives (or type II error rate).” I am not sure whether this is type I or type II error rate. Please double check and refer to https://en.wikipedia.org/wiki/Confusion_matrix.
- for → before:
The second “for” in the sentence “*We used this sample to demonstrate that self-link and heteromeric crosslinks must always be separated for FDR and that data must be merged into PPIs for correct estimation of error in crosslinking-based PPI investigations.*” had better be changed to “before” so as to remove ambiguity and to keep consistent with the sentence later: “*CSMs must be merged into PPIs before FDR estimation of PPIs (Fig. 1c).*”
- Figure-legend mismatch:
Fig. 1c shows that “CSMs 10% error” induces “PPIs 25% error”, but Fig. 1c legend is “*Schematic showing error increase when merging crosslinked residue pairs to PPIs.*” Anyway there is ambiguity.
- Reference update:
21. Leitner, A. *et al. Towards Increased Reliability, Transparency and Accessibility in Crosslinking Mass Spectrometry. arXiv [q-bio.OT] (2020).* Now it is published: Structure, Volume 28, Issue 11, 3 November 2020, Pages 1259-1268.
- Acknowledgements:
Sin-Min He → Si-Min He
- Data availability:
At present, data have not been made available, since no username plus password was provided to access the ProteomeXchange datasets.

I hope that the publication of the manuscript together with the peer review report may provide readers with a comprehensive view of the FDR control. We can always ask a question: How reliable is a reliable FDR control? I just notice that Elias and Gygi showed foresight in the title of their classic paper, "Target-decoy search strategy for increased confidence in large-scale protein identifications by mass spectrometry" (NatMeth-2007), in which they used "increased confidence" rather than "reliable".

He, Si-Min
<http://pfind.ict.ac.cn>
smhe@ict.ac.cn
 2021-3-18

Reviewer 3

I think the authors have done their best to address all of my concerns. I appreciate their efforts in addressing my three major concerns, and their gains are clear: they have gained increased confidence on their validation experiment design, on the validation software xiFDR, and on the search engine xiSearch. The manuscript can be published either as is or subject to some minor revisions as indicated below.

→ Optional discussion of possible limitation/optimization of design/analysis of the fraction-based validation method

False discovery rate (FDR) control is a cornerstone of proteomics. While the target-decoy approach (TDA) becomes the method of choice for FDR control of mass spectrometry (MS) based protein and protein crosslinking identification, the rationality of TDA-FDR control is mainly based on vague assumptions, and its validation has always been a challenge due to lack of gold benchmark methods with scalability, credibility and easy annotation. For crosslinking MS, Mechtler's validation method (NatComm-2020), based on synthetic peptides crosslinked in a small space yet searched in a large space, is both credible and easy to annotate, but it can hardly scale up to complex samples and to the protein-protein interaction (PPI) level, which is why Rappsilber's validation method proposed in this manuscript stands out: for validation of crosslinking in a complex sample, at the PPI level, together with the other three levels below, intra/inter separated and multi-level TDA-FDR control is shown to be necessary, and meantime sufficient by xiFDR + xiSearch.

The authors' optimism is shown in words in the manuscript: "In this work, we experimentally demonstrated that Crosslinking MS can reliably identify PPIs using the target-decoy approach as a quantitative error metric. This negates the need for any additional heuristics suggested by others²⁷." And also in the rebuttal: "As you can see from Figure S2 using four control types, 'non-crosslinkable', (multiple) entrapment, wrong precursor and wrong crosslinker mass, the estimation of FDR by the target-decoy approach – when done as we suggest in our work – is surprisingly accurate given the limitations listed above (proposed by the reviewer). This was a relief to us and, we assume, for the field." They are so optimistic that they actually do not discuss any limitation of their methods in Discussion.

We have added caveats of the TDA to the discussion section: "Decoys are only a model of false positives with a number of underlying assumptions¹⁶ and they cannot model false positives that do not arise from spectral matching, such as peptides non-covalently associating during LCMS²⁸. Considering these caveats, it is reassuring that our four different controls closely agree with the outcome of the target-decoy approach."

Specifically, their optimism is based on the results shown in Fig. 2e, Supp Figs 2 & 3, which are assembled as follows:

Heteromeric FDR (%) BS3 DSSO (Fig 2e, Supp Figs 2&3)	Decoy-based xiFDR	Fraction-based Non-crosslinkable	Entrapment (E.coli + Human)	Entrapment (E.coli +Yeast)	Entrapment (E.coli +Human +Yeast)	Wrong precursor mass	Wrong crosslinker mass
PPI	5.0 5.0	6.6 4.9 *	1.4 3.6	6.1 4.2	5.5 4.2	6.2 7.0	7.4 5.3
Residue Pair	5.0 5.0	5.5 4.8					
Peptide Pair	5.0 5.0	5.3 4.4					
Crosslink-Spec Match	5.0 5.0	4.8 4.2					
* 3.2 2.1 if without iBAQ cutoff							

The data in the above table show really good consistency between the FDRs estimated by the decoy-based xiFDR and the FDRs estimated by several independent methods, especially Rappsilber's fraction-based validation method. However, this consistency is subject to several approximations.

To estimate the fraction-based FDR, the key is to define non-crosslinkable or false results. By experiment design, all identified inter-fraction crosslinks are false, and all identified intra-fraction crosslinks are true.

This is the basic assumption in Mechtler's paper, but not in ours. For us, intra-fraction matches are plausible, but not necessarily correct. Methods: "While all matches in the false search space are false by definition, some matches in the plausible search space will also be random."

But in the practice of the manuscript, identified intra-fraction crosslinks may still be considered false if the associated proteins did not pass a detection limit, i.e. an iBAQ cutoff. Heuristically, the iBAQ cutoff threshold is set at an abundance of 5% or 4.3E6, and this affects the results notably: with iBAQ cutoff, the estimated PPI-level FDR is 6.6% (BS3) or 4.9% (DSSO), close to 5% TDA-FDR; without iBAQ cutoff, the estimated FDR is 3.2% (BS3) or 2.1% (DSSO), away from 5% TDA-FDR. While the iBAQ cutoff is reasonable, the choice of specific threshold is arbitrary to a certain extent; furthermore, a non-crosslinkable result is defined (identified and quantified) by MaxQuant, and another linear peptide search engine might provide different results.

We explain our considerations around the choice of cut-off in the supplement. We are glad to see the reviewer agree in our choice, and we acknowledge that the cut-off is based on a number of (reasonable) assumptions.

Similarly, for xiFDR, there is also a heuristic step of prefiltering crosslink-spectrum matches (CSMs): "Results were filtered prior to FDR to crosslinked peptide matches having a minimum of three matched fragments per peptide, a delta score of 15% of the match score and a peptide length of at least six amino acids." The choice of parameter values differs in different papers, indicating something arbitrary that may also affect the final consistency. In computational proteomics, all such approximations are understandable, and the overall consistency achieved is impressive. However, it should be confessed that in comparison, Mechtler's validation design seems simpler and more objective: all intra-group CSMs are correct, and all others are incorrect, with no additional thresholds or software involved. Therefore, at least when validating CSM- and peptide-pair-level FDRs, Mechtler's validation method seems more convincing than Rappsilber's validation method.

We prove our approach using biological material that approximates the complexity of a cell. This encompasses the natural dynamic range of protein intensities and peptide modifications. Making a sample to investigate the target-decoy approach to PPI-FDR using synthetic peptides is possible, but would be extremely time-consuming and expensive for little gain over the presented.

Even if there is perfect consistency between the FDR estimations of xiFDR-like TDA methods and other independent FDR estimation methods, is this consistent FDR equal to the true FDR? Probably not. We should be aware of the limitations of the TDA model, as I already pointed out clearly in the first-round review. Till now, routine proteome identification has no agreed TDA-FDR control at the protein level. Proteome crosslinking identification is more complex: even with state-of-the-art software xiSearch+xiFDR on the data set of the manuscript, the identification rate of tandem mass spectra is no more than 20%, as stated in the rebuttal. Therefore, we should be cautious when treating the PPI-level TDA-FDR for proteome-scale crosslinking MS. Rappsilber's validation method marks a significant step forward, but it will not negate the need for any new validation methods, including optimization of their own. xiFDR might be the best method at present to control FDR especially at the PPI level, but it will not negate the need for further validation and improvement.

FDR estimation and spectra matching procedures are rapidly evolving areas of study. We want to stress again that we have shown that our decoy matches are modelling the false matches in this dataset. However, together with this reviewer, we look forward to further improvements in the future.

→ About the Reviewer Response Figure 1 in the rebuttal

The Reviewer Response Figure 1 in the rebuttal raises intriguing concerns. Since the data set of this manuscript has not been made available yet, and details about the data analysis procedure associated with the figure have not been adequately specified, further confirmation cannot be done until the manuscript is published.

The data was made available via ProteomeXchange, and we regret that the login credentials did not make their way to the reviewers.

One highly probable reason for the higher observed CSM errors of pLink2 and MeroX compared to xiSearch+xiFDR, as shown in the figure, is that by default, pLink2 and MeroX apply TDA-FDR control to redundant CSMs, while xiSearch and xiFDR apply the control to unique CSMs. After TDA-FDR control of pLink2, if those redundant CSMs are merged into higher levels with no further TDA-FDR control, a higher FDR will be induced, as already measured either by the 15N-labeling method of pLink2 (NatComm-2019) or by Mechtler's method (NatComm-2020).

We have tested pLink 2 with Mechtler's method, and found that (1) pLink2's TDA-FDR control at the redundant CSM level was effective at the same level; (2) after merging those redundant CSMs into unique CSMs, yet without further TDA-FDR control, then the calculated FDR at the unique CSM level was higher than the calculated FDR at the redundant CSM level; (3) conducting TDA-FDR control at the unique CSM level by a new version of pLink2 was effective at this new level. In particular, all these three facts are no surprise as expected. The new version of pLink2, to be made public soon, has been equipped with TDA-FDR control at higher levels (including the unique CSM level) and has passed validation of Mechtler's method. FDR control at the redundant CSM level will still be supported in pLink2, since it is still useful, e.g. when calculating the interpretation rate of tandem mass spectra.

Therefore, if the authors construed pLink2's output of redundant CSMs (after TDA-FDR control at 5%) as unique ones, calculated the FDR by their non-crosslinkable validation method (possibly with iBAQ cutoff and normalization), and drew the figure, then the higher FDRs shown in the figure would be no surprise. The authors can easily check this. To our knowledge, CSMs reported by MeroX were also redundant.

The CSM results for both MeroX and pLink2 are redundant per default, which were the settings we used. We were aware of this and have also calculated the errors on the redundant, heteromeric CSMs, not only on the non-redundant heteromeric CSMs. We, therefore, don't see evidence for redundancy to be the reason for the high errors. Our observations hold true for the current public release of pLink2 (Jan 2020 release, version 2.3.9).

The authors indicated another hypothesis that the discrepancies between pLink2/MeroX and xiSearch shown in the figure may come from improper decoy generation, internal rescoring of target and decoy matches by machine learning, or sub-score thresholding. pLink2's TDA-FDR control at the redundant CSM level uses protein sequence reversal for decoy generation, Percolator-like re-ranking, and no sub-score thresholding (NatComm-2019). Percolator has been an established technique since its publication (NatMeth-2007), with little discussion seen yet about whether it misuses the target-decoy approach (I will contact the authors of Percolator later). In principle, Percolator will surely suppress decoy matches and meantime some (hopefully incorrect) target matches sharing similar features with the decoy ones, but some other target matches, incorrect yet with features distinct from decoy ones, may survive from the suppression.

Percolator-like approaches can suffer from matches that have a dominant true positive or dominant false positive part of a match. This can happen in standard proteomics as well (e.g. partial sequence similarity

between correct and incorrect match), but is far more likely to happen in crosslinking analysis because there are two independent peptides. It can therefore be non-trivial to detect overfitting of the machine learning model, and so any machine-learning in connection with crosslinking needs to be very carefully evaluated. The data presented in the rebuttal figure 1 would suggest this overfitting is occurring in plink2.

Anyway, decoy matches can only model certain kinds of random target matches, but cannot model all incorrect target matches, especially homologous errors, which is why I always worry about the reliability of TDA-FDR control of any kind, including xiFDR and that of pLink2. For crosslinking TDA-FDR control, one more complex issue arises: besides random-random peptide-peptide matches, there are also correct-random peptide-peptide matches, and these two kinds of incorrect matches might have different score distributions. Anyway, necessary and sufficient conditions for a search engine to be eligible to work with xiFDR or any other TDA-FDR control are worth further investigation, which is often neglected by both regular proteomics and crosslinking proteomics.

As a note regarding error in machine learning procedures: Mechtler et al. applied Kojak with PeptideProphet and saw 2.4% actual error when aiming for 5%. However, applying Kojak with Percolator yielded 22.8-31.9% actual error (see their Figure 3).

<https://www.nature.com/articles/s41467-020-14608-2>

→ About iBAQ cutoff:

Please display several best-quality CSMs that are identified by xiSearch+xiFDR but considered non-crosslinkable due to iBAQ cutoff.

Our iBAQ cut-off is sensible, as is explained in the supplement. There will be some correct matches to proteins below the cut-off. There will also be false matches to proteins above. Here are some examples for plausibly correct matches involving proteins below the chosen iBAQ cut-off:

Please show the three percentages of unique CSMs at 5% TDA-FDR that are with $iBAQ = 0$, $0 < iBAQ \leq 5\%$ ($4.3E6$), and $iBAQ > 5\%$, respectively.

We have performed this analysis and provide the results at the end of this document (Table R2).

→ About normalization factor:

Please use more words to describe the principle and the method of calculating the normalization factor 1.09 for non-crosslinkable validation experiment at the PPI level. I am sorry I cannot understand yet.

The methods have been adapted to improve clarity on this point.

→ About lower-level validation:

“Supplementary Figure 3: Observed errors using the non-crosslinkable control on information levels lower than PPI-level.” There is no definition of “observed error” in the main text or supplementary information. There is only definition of Error_PPI, non-crosslinkable in Methods, but no similar definitions for any errors using the non-crosslinkable control on information levels lower than PPI-level. Are they entirely the same as Error_PPI, non-crosslinkable, including the same normalization factor 1.09?

Yes. We changed the methods text.

→ Type I or type II error:

“This approach assumes that the rate of matches to the decoy database is an estimator of false positives (or type II error rate).” I am not sure whether this is type I or type II error rate. Please double check and refer to https://en.wikipedia.org/wiki/Confusion_matrix.

It is indeed a type I error, we have changed it in the text.

→ for → before:

The second “for” in the sentence “We used this sample to demonstrate that self-link and heteromeric crosslinks must always be separated for FDR and that data must be merged into PPIs for correct estimation of error in crosslinking-based PPI investigations.” had better be changed to “before” so as to remove ambiguity and to keep consistent with the sentence later: “CSMs must be merged into PPIs before FDR estimation of PPIs (Fig. 1c).”

We agree and changed it accordingly.

→ Figure-legend mismatch:

Fig. 1c shows that “CSMs 10% error” induces “PPIs 25% error”, but Fig. 1c legend is “Schematic showing error increase when merging crosslinked residue pairs to PPIs.” Anyway there is ambiguity.

Thanks for pointing this out - we adapted the legend to “Schematic showing error increase when merging crosslinked residue pairs to PPIs.” to increase clarity.

→ Reference update:

21. Leitner, A. et al. Towards Increased Reliability, Transparency and Accessibility in Crosslinking Mass

Spectrometry. arXiv [q-bio.OT] (2020). Now it is published: Structure, Volume 28, Issue 11, 3 November 2020, Pages 1259-1268.

We updated this reference.

→ Acknowledgements:

Sin-Min He to Si-Min He

We apologise for this mistake. We have now changed it.

→ Data availability:

At present, data have not been made available, since no username plus password was provided to access the ProteomeXchange datasets.

I hope that the publication of the manuscript together with the peer review report may provide readers with a comprehensive view of the FDR control. We can always ask a question: How reliable is a reliable FDR control? I just notice that Elias and Gygi showed foresight in the title of their classic paper, "Target-decoy search strategy for increased confidence in large-scale protein identifications by mass spectrometry" (NatMeth-2007), in which they used "increased confidence" rather than "reliable".

We thank the reviewers for their careful and considered thoughts and comments that have improved our manuscript.

Table R2: Fractions of heteromeric CSMs by iBAQ abundance of one protein of a PPI in the non-crosslinkable control

dataset	n unique CSM (5% heteromeric CSM-FDR)	fraction, iBAQ = 0	fraction, iBAQ ≤ 4.3E6	fraction, iBAQ > 4.3E6
BS3, heteromeric CSMs	3572	2.4%	2.0%	95.6%
DSSO, heteromeric CSMs	5333	2.3%	1.6%	96.1%